# Spatial Analysis Model for the Evaluation of the Territorial Adequacy of the Urban Process in Coastal Areas

Federico B. Galacho-Jiménez ⬤ and Sergio Reyes-Corredera *⬤

Geographic Analysis Group, Department of Geography, University of Málaga, Teatinos Campus, s/n, 29701 Malaga, Spain; fbgalacho@uma.es
* Correspondence: sergioreyes@uma.es

**Abstract:** Coastal spaces are shaped by human activity. Approaching their urban spaces allows us to analyse the concepts of structure, growth, and management. Highlighting the problems associated with these concepts can lead to intensive scientific analysis and provide solid research methods. This paper focuses on the study of how the process of territorial occupation takes place and the urban forms it generates on the Spanish Mediterranean coast. It is based on the consideration that the process of territorial occupation is deficient in its territorial adequacy. To analyse this, a methodology is proposed that addresses processes of analysis at different scales: dynamics of changes in land use, the study of the morphologies of urban development with spatial analysis tools, and the adaptation of urban processes to the characteristics of the spaces that support them with multi-criteria evaluation techniques and GIS (Geographical Information Systems). The results are specified in five degrees of suitability of the occupation of the territory. Two conclusions can be observed: first, urban planning gives rise to forms of occupation that follow a similar pattern in the twenty areas studied, and second, the suitability of the urban process is not governed by planning based on precepts of suitability and environmental logic but by a weakness of the adapted planning methods.

**Keywords:** land use change; urban structure; urban growth; coastal areas; GIS; multi-criteria evaluation





## 1. Introduction

The analysis of the spatial forms of occupation generated by the urban process through urban planning and territorial planning requires a special detailed analysis. In addition, it is necessary to develop methods that allow progress in the evaluation of the territorial systems that are being generated and, in turn, allow the simulation of the dynamics of change for their assessment [1–4]. This work attempts to provide a method for identifying saturated areas and defining the character that should be attributed to each of them, according to their position in the territorial model and their defining characteristics. Some authors have proposed methods along these lines [5,6]. This procedure should lead to the delimitation of areas according to their functioning and according to their level of homogeneous behaviour, areas that will be understood as elementary and significant parts of the prevailing territorial model [7–10].

The growth of cities and their surrounding areas does not always take place in accordance with the conditions of the space that supports this growth. The problem is that most planners continue to ignore what the territory expresses. The occupation of space has been developing in aggregates, by fragments from different periods related to historical and environmental processes and by the pressures of the economic agents of each time. The occupation of space in today's age is realised by bringing together real estate properties intended for different uses and served by various networks. This means that what is built on the land that serves the urban process is not a matter of planning but of the owners of this land. This means that the urban process is developing closer to dehumanisation than to a return to nature, and the living conditions of the inhabitants have been clearly

affected. For example, urban green areas are designed as complementary elements of the urban and not as elements of territorial planning, and therefore not as one of the elements that define the territorial structure. Parks, urban forests, tree-lined streets, and riverbanks support urban well-being by supplying space for rest, relaxation, and exercise, and by keeping temperatures down. However, not everyone across Europe enjoys equal access to green space in cities. Several studies have shown that the evidence of socio-economic and demographic inequalities in access to the health benefits derived from urban green and blue spaces across Europe. It highlights examples of green spaces that were designed to meet the needs of vulnerable and disadvantaged social groups [11–16].

This is based on the hypothesis, or rather, on the consideration that the process of territorial occupation that has been developed in the Spanish Mediterranean area is deficient in its adaptation to the territory where it is produced. The occupation of this geographical space has been characterised by following a disjointed development model that gives rise to a disjointed and indiscriminate urban occupation which, in addition, is a high consumer of natural resources: water, soil, natural spaces, etc. As a result, we find forms of spatial occupation that are highly inefficient in environmental terms and highly consuming of resources and infrastructures. This paper will attempt to demonstrate this fact. On the one hand, the limitations of the forms of occupation of the territory will be recognised and, on the other hand, an analysis will be made of the use of land in urban areas and their areas of influence. At the same time, it is intended to elucidate whether the mode of articulation is similar or different in the areas of the study area proposed in this work, which could be basic information for possible proposals for intervention in the framework of the territorial structure, and therefore, allowing the evaluation at this level of the consequences of land occupation actions with regard to their economic, environmental and social aspects [17,18]. Furthermore, the method aims to supply the knowledge to help design the provision of urban facilities and services based on sufficient foundations [19].

From a theoretical point of view, we place ourselves in the framework of urban growth modelling, which has proven to be a useful tool for measuring the dynamics of land use in the territory. This implies offering application possibilities in decision support systems [20,21], but also in the field of optimal location models and advanced development analysis of territorial systems. In the case of best location, the importance of its application arose from the business strategies of retail-oriented industries and the location problems that have been addressed by operations research, from the point of view of the optimisation of urban systems [22,23].

Other aspects that need to be alluded to in order to contextualise this text are the conceptual advances in the understanding of territorial systems. There is an abundance of recent literature exploring the complexity of these urban systems and the need to apply an integrated systems approach in research and practice [12,24–26].

Along the same lines is the analysis of territorial structure, with the consideration that ecological, agricultural, and urban space should be included, involving the dynamics of changes and the reorganisation of land use types to meet the functional needs of societies. The territorial structure is a systematic and integrated spatial structure and the actors involved in it should follow rules of ecological protection, the optimisation of agricultural production and the maximisation of best living conditions in cities [27].

Our intention in this text has been to work on the different scales involved in territorial development. Downscaling the analysis from the territorial to the urban implies, according to Childers et al. [28], the study of urban ecology, which has moved from ecology in and of cities to embrace an ecology for cities, along the lines set out by Theodorson in his anthology, published in 1961 [29]. For us, this statement embraces two concepts: the social and the spatial. The spatial analysis of the urban context will be developed along these lines. The spatial models of urban structure had been formulated based on the theoretical postulates of classical human ecology. In the social studies of urban space and particularly of the socio-spatial structure, different disciplines converge in the search for an explanation of the internal organisation of urban space to access a general theory of urban space.

However, it is important to clearly focus the object of study by differentiating the internal logic of urbanisation from other processes [24]. Research on urban ecology has shown that cities and urbanisation processes often modify environments, causing the ecology of cities to deteriorate [18]. According to this, the various ecological processes are based on the concept of "competition", which involves among other manifestations the struggle for space, the intensification of urban forms and the resulting changes in land use, whereby a social interpretation acquires a basic territorial dimension.

The methodological process we will follow is based on analysing urban growth models and observing the morphological configuration induced by the development of the urban process [30–33]. It is considered essential to establish a model that leads to understanding the way in which the space under study is organised and observed at different territorial scales, by defining forms or areas of defined functionality (for the study of the general form) and, in parallel, where the different forms that have arisen in the territory are located [34,35]. The areas thus classified will be analysed to clarify the relationships between them, i.e., according to the way they are articulated in the territory (study of the structure, through their position). This will lead to the definition of the structure of territorial models and their functioning [36–38]. An attempt will be made to offer a synthetic vision of them and to explain the territorial aspects that underlie them through their fundamental components [39,40]. On this basis, an interpretative model of the territory is constructed that seeks to establish, as reliably as possible, an explanatory and descriptive framework for the interventions carried out in this type of space [41]. To this end, geographic data are used in combination with geographic information systems and multi-criteria evaluation techniques to obtain sufficient information to evaluate the processes of occupation through their characterisation. We will start by identifying the units of the territory, and then go on to explain, through their understanding, the articulation between them, which will be the basis of the analysis of the spatial model, and in short, of the spatial analysis of the structure of the territory that we specify in the proposed area of study.

## 2. Materials and Methods

### 2.1. Study Area

This research and the application of the proposed method will occur in an area of clear urban and tourist dynamism, as it is possible to observe how the urban process acquires different forms and intensities, which in turn will allow us to observe the degree of building density and construction saturation in them. In Figure 1, we can see that the main criterion for selecting the areas that were to form part of this study was their geographical location on the Mediterranean coast, this areas are the Spanish provinces of Girona, Barcelona, Tarragona, Castellón, Valencia, Alicante, Murcia, Illes Balears, Murcia, Almería, Granada and Málaga. As is well-known, both in Spain and in other countries in the Mediterranean area, the majority of the population is in coastal areas.

This study area is fundamentally characterised by its high dynamics of land occupation, due to the dynamism of economic activity, with a high proportion of tourist activity. Table 1 shows the dimensions of the area studied and its grouping by province. In Spain, the province is a local entity with its own legal personality, determined by the grouping of municipalities and territorial division for the fulfilment of the activities of the State. There are fifty provinces in Spain, and this study focuses on the eleven provinces that make up the Spanish Mediterranean coastline.

The population volume reached 6,131,534 inhabitants in 2022, which represents 12.92% of the national population in 17.43% (88,175 km$^2$) of the surface area (505,944 km$^2$). Three territorial realities have been selected: one, the large metropolitan areas, represented by Girona, Barcelona, Tarragona, Palma, Castellón, Valencia, Alicante, Murcia, Granada, Almería, and Málaga. Another is the suburban cities, and tourist areas; these are municipalities with more than 100,000 inhabitants, with their own dynamics, which allow them to generate a continuous or discontinuous urban fabric and the functionality of a supra-

municipal area: Benidorm, Gandía, Marbella and Torrevieja. Also, there is a third typology, called modern and complex suburban spaces: Manresa, Cartagena, Elche-Santa Pola, Elda, and Reus. The characteristics by which they have been defined are explained below.

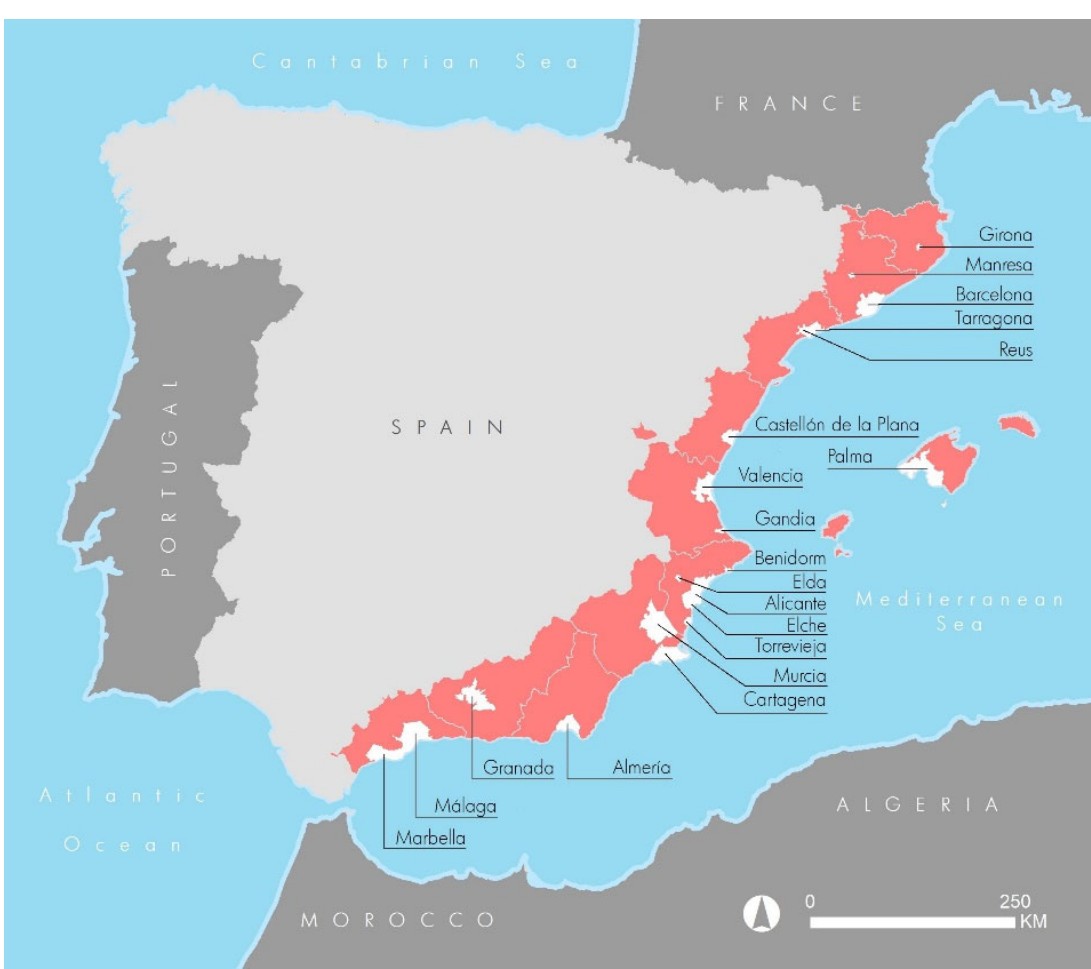

**Figure 1.** Location map of the study area.

**Table 1.** Population and surface characteristics of the studied areas that are part of the study area. Geographical arrangement from north to south.

| Provinces | Areas | Inhabitants 1991 | Inhabitants 2022 | % Increase | Surface (km²) | % Surf./Prov. | Density 1990 (Inh./km²) | Density 2022 (Inh./km²) |
|---|---|---|---|---|---|---|---|---|
| GIRONA | Girona | 68,656 | 102,666 | 49.54 | 39 | 0.04 | 1754.11 | 2623.05 |
| BARCELONA | Manresa | 66,320 | 77,452 | 16.79 | 42 | 0.05 | 1591.93 | 1859.15 |
| | Barcelona | 1,643,542 | 1,636,193 | −0.45 | 101 | 0.11 | 16,216.50 | 16,143.99 |
| TARRAGONA | Tarragona | 110,153 | 134,883 | 22.45 | 63 | 0.07 | 1748.46 | 2141.00 |
| | Reus | 87,670 | 106,741 | 21.75 | 53 | 0.06 | 1652.59 | 2012.08 |
| PALMA | Palma | 296,754 | 415,940 | 40.16 | 209 | 0.24 | 1422.39 | 1993.67 |
| CASTELLÓN | Castellón | 134,213 | 171,857 | 28.05 | 111 | 0.13 | 1205.54 | 1543.67 |
| VALENCIA | Valencia | 752,909 | 792,492 | 5.26 | 135 | 0.15 | 5591.60 | 5885.57 |
| | Gandía | 51,806 | 75,911 | 46.53 | 61 | 0.07 | 852.07 | 1248.54 |

**Table 1.** *Cont.*

| Provinces | Areas | Inhabitants 1991 | Inhabitants 2022 | % Increase | Surface (km²) | % Surf./Prov. | Density 1990 (Inh./km²) | Density 2022 (Inh./km²) |
|---|---|---|---|---|---|---|---|---|
| ALICANTE | Benidorm | 42,442 | 69,738 | 64.31 | 39 | 0.04 | 1102.10 | 1810.91 |
| | Elda | 54,350 | 52,297 | −3.78 | 46 | 0.05 | 1186.94 | 1142.11 |
| | Alicante | 265,473 | 338,577 | 27.54 | 201 | 0.23 | 1318.99 | 1682.20 |
| | Elche–Santa Pola | 188,062 | 235,580 | 25.27 | 326 | 0.37 | 576.75 | 722.48 |
| | Torrevieja | 25,014 | 83,547 | 234.00 | 71 | 0.08 | 350.14 | 1169.47 |
| MURCIA | Murcia | 328,100 | 462,979 | 41.11 | 882 | 1.00 | 372.05 | 525.00 |
| | Cartagena | 168,023 | 216,961 | 29.13 | 558 | 0.63 | 301.07 | 388.76 |
| ALMERÍA | Almería | 155,120 | 199,237 | 28.44 | 296 | 0.34 | 524.92 | 674.21 |
| GRANADA | Granada | 255,212 | 228,682 | −10.40 | 88 | 0.10 | 2899.48 | 2598.07 |
| MÁLAGA | Málaga | 522,108 | 579,076 | 10.91 | 395 | 0.45 | 1321.86 | 1466.09 |
| | Marbella | 80,599 | 150,725 | 87.01 | 117 | 0.13 | 688.17 | 1286.93 |
| | Total, Areas | 5,296,526 | 6,131,534 | 15.77 | 3832 | 4.35 | 1382.09 | 1599.98 |
| | Total, Provinces | 13,724,866 | 18,537,697 | 35.07 | 88,175 | 100 | 155.65 | 210.24 |

Data source: National Statistics Institute. Spanish Statistical Office.

*2.2. Data Sources*

Four data sources were used for this work, selected because of their usefulness for the proposed spatial analysis at different scales.

Firstly, an analysis of the dynamics of changes in land use in the provinces between 1990 and 2018 was undertaken. The aim was to observe the transformations in land use in these areas, which fundamentally describe the dynamics of changes in urban use and related uses: industrial areas, infrastructures, and commercial areas. For this purpose, the map of land occupation in Spain, scale 1:100,000, corresponding to the European Corine Land Cover project (CLC), was used as a source. The 1990 and 2018 versions available were used. The intention was to cover a period of 30 years, which was considered significant for the analysis of these changes.

Secondly, for the analysis of the territorial occupation by the urban process in the study area and to delimit the areas considered most significant to be studied, the Geographic Information Reference Populations database, version 2020.1, IGR Populations Project (IGRPP) was used. This is a spatial dataset that is designed to provide the geographical location and geometric shape of the entities and population areas, with coverage of the Spanish territory at a scale of 1:5000. The date of the data is 2017. It was used to interpret land occupation by means of the geometric shape of the populations. Populations are understood as those groupings of more than one building and its associated spaces that are known by the same name, including those of residential use, infrastructures, and industrial areas. These groupings of buildings are geometrically defined on the cadastral parcel, so that it was possible to relate this source of data with the others used in this work. As a result, the areas with the highest concentration of urban land with its different typologies were selected, thus making it possible to advance in the forms of special occupation. Thirdly, a large amount of land cover data is concentrated in the Copernicus Land Monitoring Service. From this source, we used the product "Urban Atlas" (URBAN COVER) from the Copernicus "Urban Atlas" project for the reference year 2006, the "Urban Atlas" update and extension for the reference year 2012, and the High-Resolution Land Occupation Information System in Spain (SIOSE HR) for the year 2017. This is a SIOSE land occupation database for the whole of Spain at a scale of 1:25,000, with a reference date of 2005 and updates after the reference years 2009, 2011, 2014 and 2017.

Table 2 shows the correspondences between the values of the three data sources mentioned above and the adjustments that have been made to the data model. The denominations that appear will be the ones that will be used throughout the work.

**Table 2.** Data model. Nomenclatures and correspondences between the data sources used: Corine Land Cover (CLC), Geographic Population Reference Information (IGRPP) and Urban Atlas (URBAN COVER) with High Resolution Land Occupation Information System in Spain (SIOSE HR).

| USES | CLC | IGRPP | URBAN COVER—SIOSE HR | UC |
|---|---|---|---|---|
| ARTIFICIAL SURFACES | Continuous urban fabric | Residential | CUF: Continuous urban fabric (>80%) | (1) |
| | Discontinuous urban fabric | Built-up areas | DUF (A): Discontinuous dense urban fabric (50–80%) | (2A) |
| | | | DUF (B): Discontinuous medium density urban fabric (30–50%) | (2B) |
| | | | DUF (C): Discontinuous low density urban fabric (10–30%) | (2C) |
| | | | DUF (D): Discontinuous very low density urban fabric (<10%) and/or isolated structures, construction sites and land without current use | (2D) |
| | Industrial or commercial areas | Industrial | ICT: Industrial, commercial, public, military, and private units | |
| | Roads, railways, and associated land networks | Infrastructure | ICT: Fast transit roads and associated land | (3) |
| | | | ICT: Other roads and associated land | |
| | | | ICT: Railways and associated land | |
| | | | ICT: Port areas | |
| | | | ICT: Airports | |
| | Mining extraction areas | Abandonment | MAL: Mineral extraction and dump sites | (4) |
| | Urban green areas | Green zone | AGA: Green urban areas | (5) |
| | Sports facilities | Service | AGA: Sports and leisure facilities | |
| AGRICULTURAL | Heterogeneous agricultural areas | Primary | Arable land (annual crops) | (6) |
| | Annual crops | | AAZ: Permanent crops (vineyards, fruit trees, olive groves) | |
| FORESTAL | Natural grasslands | | NVA: Pastures | (7) |
| | Forests | | NVA: Forests | |
| | Shrub and herbaceous | | NVA: Herbaceous vegetation associations | |
| | Open spaces | | OSV: Open spaces with little or no vegetation | (8) |
| WETLANDS | Coastal wetlands | | WWZ: Wetlands | (9) |
| | Water sheets | | WWZ: Water | |

Data source: Those cited above.

In order to characterise the forms of urban occupation, other data sources were used to obtain the necessary parameters for the multi-criteria evaluation analysis that will be presented in the methodology section. The population data available from the Spanish National Institute of Statistics from 2023 were used, with reference to the census section, i.e., the most detailed official geographical reference for carrying out demographic studies in Spain. The population density parameters set by the Changing Mediterranean Metropolises Around Time (CAT-MED) project were used to establish a measurement indicator.

The study of the distribution and density of dwellings was based on data available from the National Institute of Statistics and the density parameters available in the Special Plan for Environmental Sustainability Indicators of Urban Development Activity in the city of Seville were followed.

The study of the number of square metres of green spaces per inhabitant was based on the recommendations of the United Nations used in the CAT-MED project and used the available data on green spaces from the National Geographic Information Centre of the National Geographic Institute for the year 2023.

Based on the aforementioned data sources, the following methodological process has been carried out, seen in Figure 2:

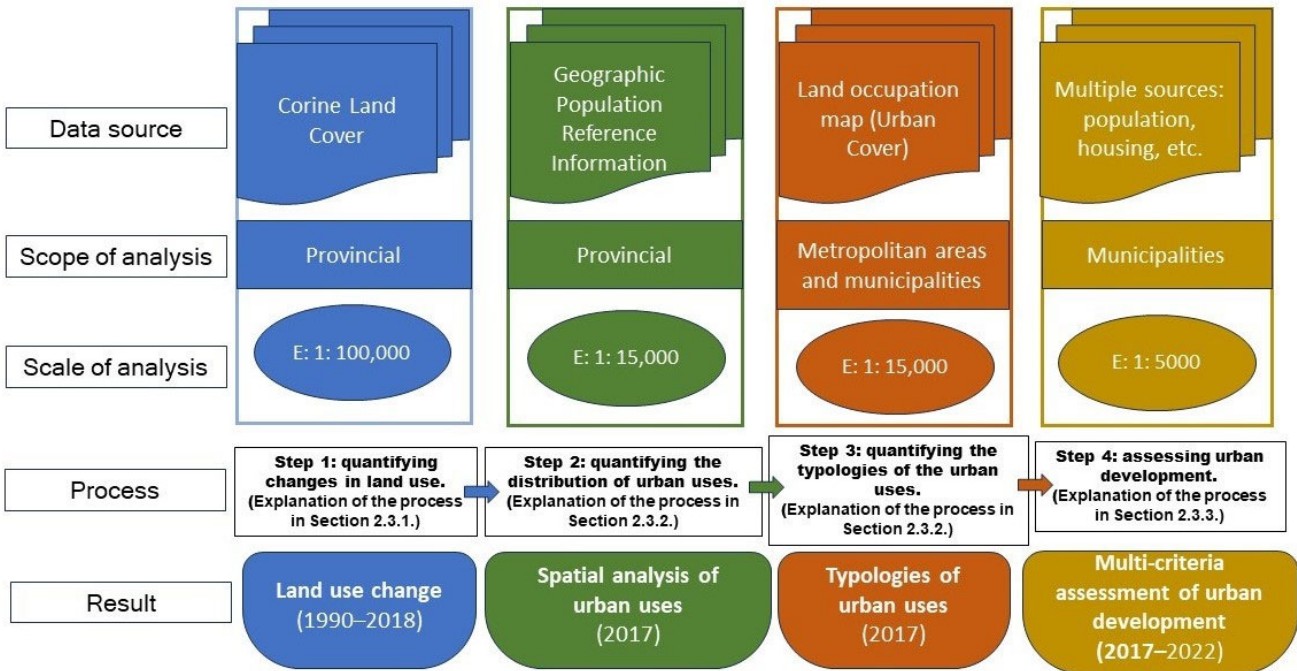

**Figure 2.** Analysis process and data sources.

*2.3. Metodology*

2.3.1. Analysis of the Dynamics of Land Use Change

This process corresponds to the first step of the methodology. It addresses the analysis of land use change patterns and their spatial distribution. We rely on the change detection methodology developed by Pontius et al. (2004) [42] and Pontius and Malizia (2004) [43], which has been used by numerous authors for the quantitative analysis of land use change dynamics [44–48]. Our objective, in this case, is to show the most dynamic spaces, with respect to the changes in uses related to the urban process in the study area, and to quantify the surface areas of these changes. The analysis process begins with the establishment of the temporal criteria for analysis, which in this work are the years 1990 and 2018. As we have the land use information layers already prepared for these periods, we proceed directly to the analysis of changes, which is based on the superposition of the land use layers for the established dates. Using the geoprocessing tools offered by ArcGIS 10 and ArcGIS Pro, we carry out the geometric intersection of the two layers and create a new layer that computes the coincidences of both layers and the attributes associated with them. The results of the operation are transferred to a cross-tabulation or transition matrix for the period. This matrix lists the classes of time slice one (1990) in the rows and the classes of time slice two (2018) in the columns. On the diagonal of the matrix are, quantified by use classes, the areas that have remained stable between the two time slices, which will be labelled as *Persistences (Pjj)*, showing each usage type that persists in the type *j*. The surfaces of these same classes that show transitions to other use classes on the reference dates are located outside the diagonal and are called *Transitions (Pij)*, showing the surfaces that have undergone a transition from class *i* to class *j*, i.e., from one use class to another. These transitions can be of two types: losses and gains.

In addition, they can be multiple and depend on the number of classes from which they started. This is related to the levels of disaggregation of the data model that were established at the time of analysis. In our case, nine classes were established: (1) CUF: continuous urban fabric, (2) DUF: discontinuous urban fabric and areas under construction, (3) ICT: industrial, commercial, and transport zones, (4) MAL: mining areas and landfills, (5) AGA: artificial, non-agricultural green areas, (6) AAZ: farmland, permanent crops, and heterogeneous agricultural areas, (7) NVA: areas of natural vegetation (forests) and areas of shrub and/or herbaceous vegetation, (8) OSV: open spaces with scarce or no vegetation, and (9) WWS: continental or coastal wetlands and water surfaces. Table 2 shows the correspondences of these classes with the other sources used.

According to the definition made above, we can define the losses (from 1990 to 2018) on a given class j as the difference between the sum of row j of the matrix P (Pj*) and the diagonal value on that row, where the persistence is located (Pjj), representing the surface of each class j that experiences net losses in the period studied, expressed as: Lj = Pj* − Pjj. On the other hand, gains are the difference between the sum of column j of the matrix (P*j) and the diagonal value on that row, where the persistence is located (Pjj), expressed as: Gj = P*j − Pjj. Once we have defined losses and earnings, we define the total change for each use class j as the total surface changing its use form or to class j, expressed as: TCj = Lj + Gj. Finally, the net change in use class j is defined as the total surface in class j in 2018 minus the total surface of class j in 1990, expressed as: NCj = P*j − Pj*. These results can be found in Tables 4 and 5, which we will comment on in Section 3.1.

2.3.2. Analysis of the Spatial Structure and Quantification of the Form of the Urban Process

This analysis is carried out by combining three sources of data. On the one hand, we use the Geographic Information of Reference Populations data source (IGRPP). This data set is used to obtain the geographical location and geometric shape of the building grouping areas, as well as to interpret the data by situating them in each established area. The use of the groupings of buildings by population makes it possible to constitute areas of occupation, so that this information will later be related to the density of dwellings or inhabitants, which we will use to make the multi-criteria evaluation. On the other hand, in addition to the presence of buildings, land use is used, so that the relationship between the shapes of the land groupings and their uses is established.

Furthermore, once the areas with the highest concentration of urban uses and the morphologies that these acquire in the territory are known, we proceed to differentiate the types of occupation of these soils. To help us in this process, two other data sources have been combined with the previous one: the product "Urban Atlas" (URBAN COVER) from the Copernicus "Urban Atlas" update and extension for the 2012 reference year and the High-Resolution Land Occupation Information System in Spain (SIOSE HR) from the year 2017. With the elaboration of this information, it has been possible to show the prevailing territorial occupation model in each of the studied areas and to quantify with GIS the surfaces, according to the established classes, which are mentioned in Table 2.

However, at this point, have considered carrying out the analysis by differentiating class (2), DUF: discontinuous urban fabric and areas under construction, which has been subdivided into four subclasses, to analyse this type of territorial occupation in more detail. The established classes are: (2A) DUF (A): discontinuous dense urban fabric (50–80%), (2B) DUF (B): discontinuous medium density urban fabric (30–50%), (2C) DUF (C): discontinuous low density urban fabric (10–30%), and (2D) DUF (D): discontinuous very low-density urban fabric (<10%), isolated structures, construction sites, and land without current use.

The spatial structure is further characterised according to three parameters: the density of the inhabitants, the density of the dwellings, and the availability of green areas per inhabitant, which describe the characteristics of occupation in the different urban fabrics. A series of geoprocesses were carried out to assimilate these values to the administrative scale closest to the available population, i.e., the census section. For the density of inhabitants, the

population was related to the surface area of the census section in square metres, although it has subsequently been expressed in hectares. In the case of the density of dwellings, a similar operation was carried out, as the housing census data are available at the section level. For the preparation of the parameter for the availability of green areas per inhabitant, it was necessary to carry out an operation of superimposition and calculation of green areas, to relate them to each of the census sections and the population registered in them. In this way, it was possible to measure the number of inhabitants per square metre of green areas. These three parameters make it possible to evaluate the characteristics acquired by the different developments and the valuation of the urban fabric generated in a territory of high urban development activity, such as the Spanish Mediterranean strip.

The results of the methodological process carried out in this section are presented in Tables 6 and 7, which we will comment on in Section 3.2.

### 2.3.3. Assessment of the Territorial Adequacy of the Urban Process with Multi-Criteria Decision Analysis (MCDA) and Geographic Information Systems Techniques

The evaluation of the suitability of the implementation of urban uses in the territory is carried out through the use of multi-criteria analysis techniques and, fundamentally, according to specific objectives, which were expressed above in the Introduction section. The expected result of this process is to derive as precise a measure as possible of the suitability of the implementation of urban uses in their different forms in the spaces studied. There are a number of methods for doing this that differ more in their operational procedures than in their conceptual bases. In this work, we have chosen to apply multi-criteria evaluation techniques. The procedure applied by Barredo and Gómez [49] is followed, which is based on the contributions of various authors who have provided the scientific foundations for these procedures and techniques [50–53]. The evaluation process starts with the selection of the factors to be included in the evaluation and the scoring of their values. Four factors have been selected: (F1) land use typology, (F2) density of inhabitants (inhabitants/ha), (F3) density of dwellings (number of dwellings/ha), and (F4) area of urban green spaces (m$^2$ of green spaces/inhabitant). We believe that the best organisation to represent the relationship of the above factors and the alternatives defining their assessment is a matrix. In such a matrix, the criteria (j) can occupy the main column and the alternatives (i) the main row. We will call this matrix a matrix of scores, since the internal values of this matrix are called criteria scores (Xij), and represent the level of desirability that is established for each alternative in each factor. Once a variable is assigned weights or scores it becomes an evaluation criterion.

Once these factors have been configured and, prior to the establishment of preferences to convert them into evaluation criteria, the values of each factor have been grouped into intervals. This has been undertaken because, given the number of values that enter the scoring process, it is easier to interpret the results in a grouped way than in a disaggregated way (although, in the databases, these are kept disaggregated in case we would like to reconsider the groupings made). A Likert scale is used for this purpose, so that we can evaluate them qualitatively [54]. The classification consists of nine class intervals in which, for each of the factors, a description of the state of preference is assigned according to the values they contain. For the weighting of the values, we propose Saaty's pairwise comparison method [55]. With this procedure, a matrix is established, in whose rows and columns the number of attributes of each factor (classes) to be weighted is defined. The result is a comparison matrix between pairs of classes, in which the importance of each of them over each of the others (aij) is observed. The measurement scale established for the assignment of the value judgments (aij) is a continuous scale (ratios), ranging from a minimum value of 1/9 to nine, with extremely more adequate (1/9) to extremely less adequate (9), with the value of one indicating equality in importance between pairs of factors.

Table 3 shows the groupings of class intervals and the results of the application of Saaty's method, according to the preference criterion established by the authors of

this work, which appears in the value column. The class intervals were made by the statistical application of the mean and standard deviation, with an interval size of ½ Std Dev (Standard Deviation).

**Table 3.** Assignment of weights to the values of each of the factors, according to land use suitability preferences. Factors: F1, land use typology; F2, density of inhabitants (inhabitants/ha); F3, density of dwellings (number of dwellings/ha); and F4, surface area of urban green areas (m$^2$ of green areas/inhabitant).

| SUITABILITY | VALUE (*) | F1 | F2 | F3 | F4 |
|---|---|---|---|---|---|
| EXTREMELY LOW | 1 | (1), (4) | 876.371–1531.15 | 408.231–707.58 | 0–11.59 |
| VERY LOW | 0.723 | (3) | 747.441–876.37 | 348.661–408.23 | 11.591–24.37 |
| LOW | 0.516 | (2A) | 636.941–747.44 | 297.601–348.66 | 24.371–37.14 |
| MEDIUM LOW | 0.360 | (2B) | 526.431–636.94 | 246.531–297.60 | 37.141–49.92 |
| MEDIUM | 0.240 | (2C) | 415.921–526.43 | 195.471–246.53 | 49.921–62.70 |
| MEDIUM HIGH | 0.148 | (2D) | 305.421–415.92 | 144.401–195.47 | 62.701–75.47 |
| HIGH | 0.080 | (5) | 194.911–305.42 | 93.331–144.40 | 75.471–88.25 |
| VERY HIGH | 0.032 | (6) | 84.411–194.91 | 42.271–93.33 | 88.251–101.03 |
| EXTREMELY HIGH | 0 | (7), (8), (9) | 0–84.41 | 0–42.27 | 101.031–2535.89 |

Suitability criteria: values from 0, as the best valued, to 1, as the worst rated. (*) The value column shows the score given to the values contained in the factors. As can be seen, they are the same for all the factors and only acquire a different meaning depending on the differentiation of the intervals representing the values in F2, F3, and F4. Legend F1: (1) CUF: continuous urban fabric (>80%), (2A) DUF (A): discontinuous dense urban fabric (50–80%), (2B) DUF (B): discontinuous medium density urban fabric (30–50%), (2C) DUF (C): discontinuous low density urban fabric (10–30%), (2D) DUF (D): discontinuous very low density urban fabric (<10%), isolated structures, construction sites and land without current use, (3) ICT: industrial, commercial, and transport zones (industrial, commercial, public, military, private units, transit roads and associated land, port areas, and airports), (4) MAL: mining, landfill, and construction areas, (5) AGA: artificial, non-agricultural green areas, (6) AAZ: agricultural areas, farmland, permanent crops, and heterogeneous agricultural areas, (7) NVA: areas with natural vegetation (areas of natural vegetation (forests) or areas of shrub and/or herbaceous vegetation), (8) OSV: open spaces with scarce or no vegetation, (9) WWS: continental or coastal wetlands and water surfaces.

Once the weights have been established, decision rules are applied for the combination of the factors, to create the final suitability criterion. These rules refer to specific aspects, such as the measurement of the attributes, to give value to the final criterion, or how to integrate the factors in the evaluation of the alternatives. The logic of the evaluation process is formalized through a series of arithmetic-statistical procedures that make possible the integration of the criteria, proved in a simple composition index, providing the way to compare the alternatives using this index [56,57].

According to the value judgments considered, weights will be proposed. For example, if we have tried to give priority to the type of land occupation (e.g., F1), we have given more weight to this factor (e.g., 0.5). Secondly, it was valued as important for the definition of the suitability model that the inhabitants of the urban spaces have enough green areas (F4), so this factor was given priority with 0.35, and, finally, the factors F2 and F3, density of inhabitants and density of dwellings, respectively, were given 0.15. The ideal point assumes the value of one is the maximum of all criteria and zero is the minimum. However, it is important to note that, depending on how these weights are applied to each of the factors, the cartographic results may differ, and the resulting layer may show one aspect or another. In the weighting that we have carried out, we have considered that the forms of occupation of the territory are a priority for us, in such a way that it is considered that this occupation ensures the best conditions for the occupation of the territory by urban uses or uses related to them.

Another clarification we would like to make is that, in the multi-criteria evaluation procedure for the calculation of the ideal point, we have used compensatory techniques among the factors. We have chosen to propose these because there may be compensatory

operations in relation to them. In this way, the evaluation model will combine the possible alternatives, and the compensatory procedures are better adapted to the logic of the proposed model. This logic responds to the fact that the factors used determine an evident complementarity by their own configuration; for example, where there is a continuous urban fabric in the territory, there is a correlation with a higher density of inhabitants and housing, and at the same time, low values of green areas per inhabitant are penalized in these areas. This is also why, within the compensatory procedures, we have the possibility of using the calculation method to derive the linear ranking of the alternatives from the scores assigned to the varied factors of the distance to the ideal point. This serves to derive a single score from the scores given for the four selected factors. Consequently, the decision rule adopted, that of the ideal point analysis, considers that each criterion is represented on an axis of the multivariate space, the ideal point here being an ideal alternative that represents the maximum level of aspiration (in this case, the maximum suitability) or that offers the best possibility of selection. In this way, the deviations of each alternative are not calculated with an ideal point that must be considered unattainable, since territorial occupation has long since ceased to be a perfect model. Finally, the distance between each alternative and the ideal point is calculated so that the alternatives closest to the ideal point can be selected. The equation used was the following, taken from Gómez and Barredo [49]:

$$Lp = \left[ \sum_{j=1}^{n} w_j \left| x_{ij} - 1 \right|^p \right]^{1/p}$$

where

$w_j$ is the weight of criteria *j*, $x_{ij}$ is the value of alternative *i* in criterion *j*, and *p* is the metric for the calculation of the distance (*p* = 2 corresponds to the Euclidean distance).

### 3. Results

*3.1. Results of the Analysis of Land Use Change Dynamics*

The results obtained from the application of the procedure explained in Section 2.3.1. show the changes brought about by the evolution of the urban process that has taken place in Spain during the last thirty years. The dynamics of changes show, at the general level of the area, that 79.58% of the surfaces (7028,551 ha) have remained in the same uses they had in 1990, during the period of time analysed (1990–2018), on a total surface of 8,831,807 ha. Therefore, the areas that have changed use account for 20.42% (1,803,256 ha). Bearing in mind that we make this comment at the provincial level, where the areas studied are located, the breakdown of the figures for changes is as follows: 0.37% of the surfaces (32,925 ha) have become continuous urban fabric, 1.49% (132,013 ha) have become discontinuous urban fabric, 0.91% (80,721 ha) have changed to commercial, industrial, and infrastructure uses and, finally, changes in agricultural uses have accounted for 17.64% (1,557,597 ha). The assessment of these figures must be made according to the scale of analysis we have mentioned, and this means that many of the areas that do not change are forested areas and areas with consolidated or permanent agricultural crops. However, the number of changes is concentrated in the areas where the urban process is most intense, as we will see below. Before the detailed analysis, we would like to show the global characterization of the changes according to land use, which is shown in Table 4. For this purpose, the quantification of the areas that have changed land use was transferred to a transition matrix or cross-reference table P.

**Table 4.** Matrix of the dynamic of changes in land use between 1990 and 2018 for the study area. Area in has (hectares).

| | 1990 | 2018 | LOSSES | EARNINGS | TOTAL CHANGE | NET CHANGE |
|---|---|---|---|---|---|---|
| USES | $T1\,(Pj^*)$ | $T2\,(P^*j)$ | $Lj = Pj^* - Pjj$ | $Gj = P^*j - Pjj$ | $TCj = Lj + Gj$ | $NCj = P^*j - Pj^*$ |
| (1) CUF | 92,528.05 | 84,912.36 | 51,989.54 | 44,373.84 | 96,363.38 | −7615.69 |
| (2) DUF | 113,570.05 | 205,668.50 | 74,281.99 | 166,380.44 | 240,662.42 | 92,098.45 |
| (3) ICT | 34,527.16 | 107,603.63 | 26,722.55 | 99,799.03 | 126,521.57 | 73,076.48 |
| (4) MAL | 15,027.89 | 32,323.29 | 11,463.86 | 28,759.26 | 40,223.12 | 17,295.39 |
| (5) AGA | 4398.48 | 24,649.05 | 3038.53 | 23,289.09 | 26,327.62 | 20,250.56 |
| (6) AAZ | 5,218,406.18 | 4,736,283.75 | 3,095,275.65 | 2,613,153.22 | 5,708,428.87 | −482,122.43 |
| (7) NVA | 4,431,961.79 | 4,975,842.51 | 3,338,846.62 | 3,882,727.34 | 7,221,573.96 | 543,880.71 |
| (8) OSV | 651,149.86 | 238,058.30 | 143,641.91 | −269,449.65 | −125,807.74 | −413,091.56 |
| (9) WWS | 542,564.77 | 232,174.56 | 532,335.19 | 221,944.98 | 754,280.17 | −310,390.21 |

Legend: (1) CUF: continuous urban fabric, (2) DUF: discontinuous urban fabric and areas under construction, (3) ICT: industrial, commercial, and transport zones, (4) MAL: mining areas and landfills, (5) AGA: artificial, non-agricultural green areas, (6) AAZ: farmland, permanent crops, and heterogeneous agricultural areas, (7) NVA: areas of natural vegetation (forests) and areas of shrub and/or herbaceous vegetation, (8) OSV: open spaces with scarce or no vegetation, (9) WWS: continental or coastal wetlands and water surfaces. Data source: Corine Land Cover (CLC). Map of land occupation in Spain, scale 1:100,000, corresponding to the European project Corine Land Cover, versions of 1990 and 2018. National Geographic Institute of Spain. Government of Spain.

In Table 4, starting from the quantified areas of each use, we observe that the exceptional creation of urban land generated by the urban development expectations that occurred in the 1990s slowed down, leaving a lot of use in urban development expectations or in a slower urban development dynamic that is accounted for in the discontinuous urban ejidos. The belief that there was little urbanized land to sustain the urban development model prevailing at that time led to the creation of huge pockets of such land, which are finally being built, as the land market demands it. The enormous economic costs derived from that model of urban land creation gave rise to another process of land occupation, which has been the creation of urban fabric in dis-continuity. This phenomenon can be observed in the rows of such uses: (1) CUF (continuous urban fabric) and (2) DUF (discontinuous urban fabric and areas under construction). Meanwhile, while the former has suffered a decrease in surface area, quantified at −7615.69 ha, the latter has increased its surface area by 92,098.45 ha. The transitions between losses and gains can be observed in the corresponding columns. Other notable changes occurred for use (6) AAZ: farmland, permanent crops, and heterogeneous agricultural areas. The losses, which show a net change of around −482,122.43 ha, are mainly due to a tendency to abandon agricultural activity in areas that are expected to be urbanized, where agriculture cannot compete with the land prices imposed by the urban process. Finally, the losses recorded for uses (8) OSV: open spaces with scarce or no vegetation and (9) WWS: continental or coastal wetlands and water surfaces respond to causes related to environmental deterioration over the last thirty years, and to issues related to climate change, in the latter, which has led to the drying up of a large number of areas previously occupied by wetlands.

On the other hand, the total change column shows the persistence of the dynamics of changes in uses and transitions between uses. Logically, these quantities are determined by the dynamism of uses such as the urban type and the persistence of uses such as the more permanent ones, like (7) NVA: areas of natural vegetation (forests) and areas of shrub and/or herbaceous vegetation, which are more affected by forest fires than by the urban process, since most of these areas are protected or cannot be urbanized according to different land laws.

The dynamics of land use changes are shown in their geographic location in Figure 3. This figure provides an overview of the changes in the field of study, so that the most

dynamic areas can be observed. Logically, these areas correspond to large metropolitan areas in the Mediterranean environment: Barcelona, Valencia, Alicante, Granada, and Malaga. The current result of the changes can be seen in more detail in Figures 4–8.

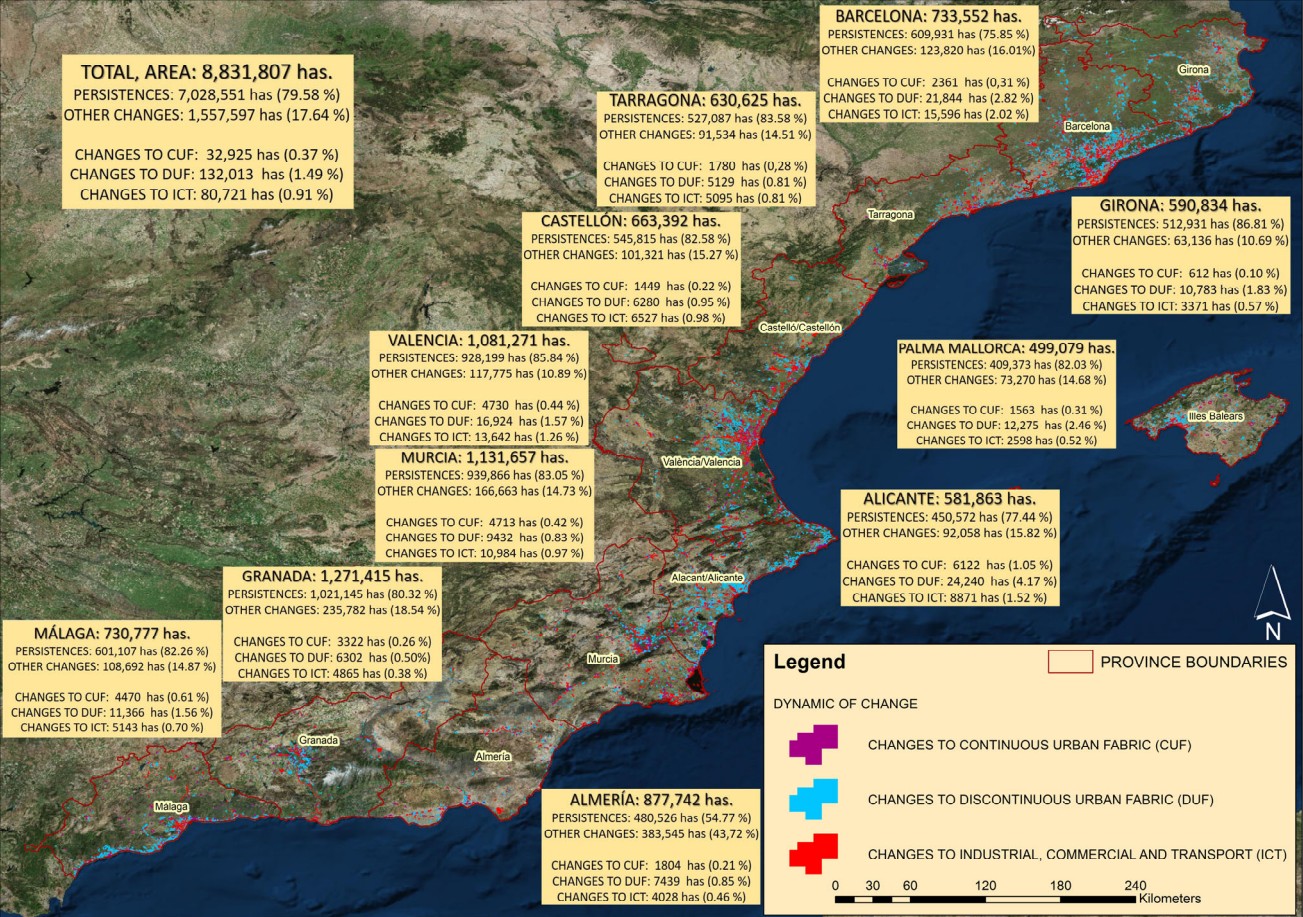

**Figure 3.** Land use change map of the study area.

A quantification has been made at the provincial level, in which we can observe several aspects. On the one hand, the persistence of uses is above 75% in all cases, except in the province of Almería, which shows a particularity in terms of transitions to non-urban uses, since the dedication of uses related to agricultural activity is very strong, and therefore, its dynamics of change are more powerful than that of urban uses. This can be seen in the value of persistences, which is 54.77%, well below the rest of the provinces, and by the value of the figure for changes in other non-urban uses (see the other changes concept for Almería), which is quantified at 43.72%.

Table 5 shows the transitions from each type of use to urban uses, which will be discussed in detail below. First, it should be remembered that in the diagonal of the matrix are, quantified by class of use, the areas that have remained stable between the two time slices (1990–2018), which will receive the denomination of persistences (*Pjj*). Outside the diagonal are the surfaces of the same classes that have transitioned to other use classes on the reference dates, which will be called transitions (*Pjj*), showing the surfaces that have undergone a transition from class i to class j, i.e., from one use class to another. These transitions can be of two types: losses (in the rows) and gains (in the columns).

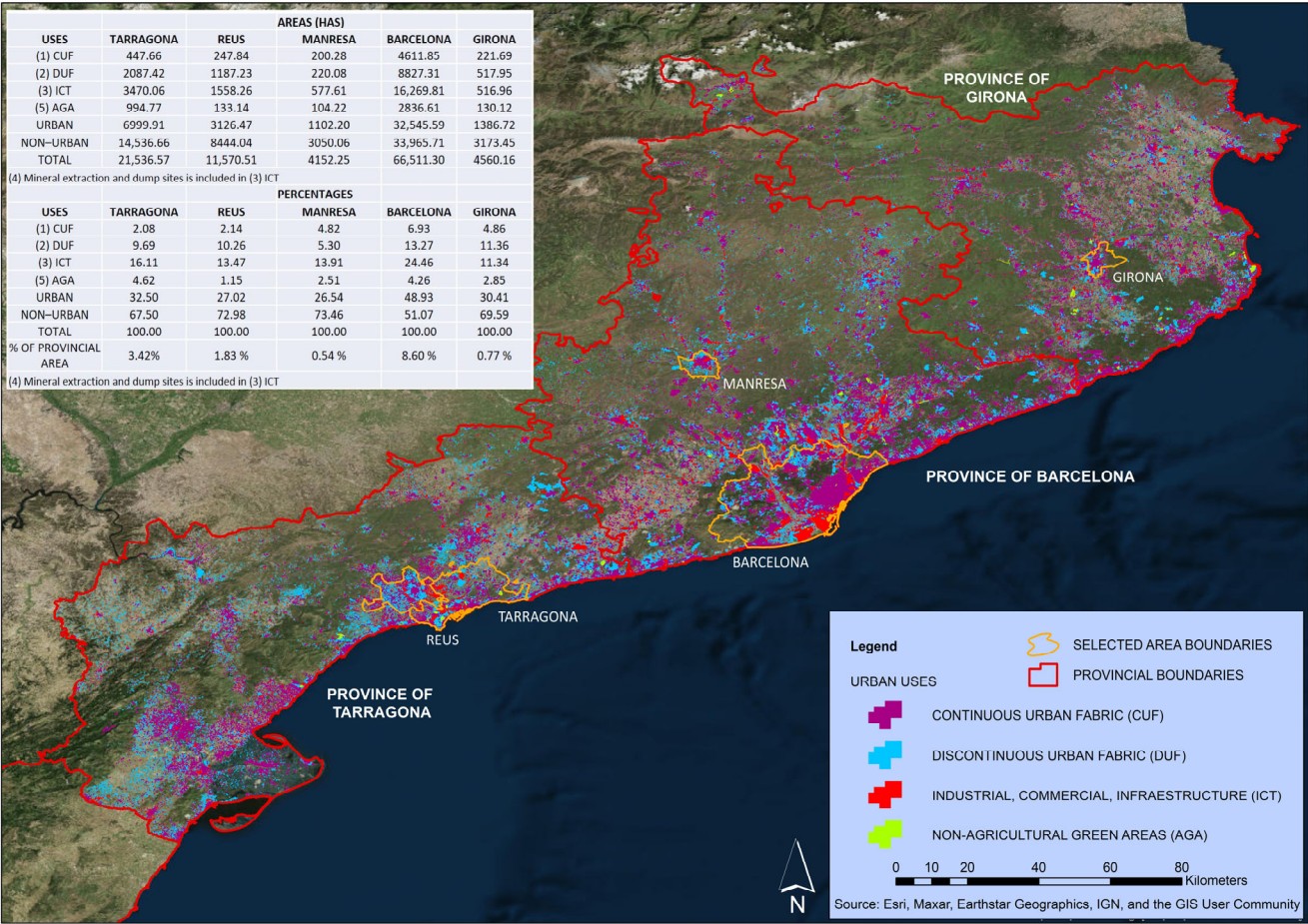

| | AREAS (HAS) | | | | |
|---|---|---|---|---|---|
| USES | TARRAGONA | REUS | MANRESA | BARCELONA | GIRONA |
| (1) CUF | 447.66 | 247.84 | 200.28 | 4611.85 | 221.69 |
| (2) DUF | 2087.42 | 1187.23 | 220.08 | 8827.31 | 517.95 |
| (3) ICT | 3470.06 | 1558.26 | 577.61 | 16,269.81 | 516.96 |
| (5) AGA | 994.77 | 133.14 | 104.22 | 2836.61 | 130.12 |
| URBAN | 6999.91 | 3126.47 | 1102.20 | 32,545.59 | 1386.72 |
| NON–URBAN | 14,536.66 | 8444.04 | 3050.06 | 33,965.71 | 3173.45 |
| TOTAL | 21,536.57 | 11,570.51 | 4152.25 | 66,511.30 | 4560.16 |

(4) Mineral extraction and dump sites is included in (3) ICT

| | PERCENTAGES | | | | |
|---|---|---|---|---|---|
| USES | TARRAGONA | REUS | MANRESA | BARCELONA | GIRONA |
| (1) CUF | 2.08 | 2.14 | 4.82 | 6.93 | 4.86 |
| (2) DUF | 9.69 | 10.26 | 5.30 | 13.27 | 11.36 |
| (3) ICT | 16.11 | 13.47 | 13.91 | 24.46 | 11.34 |
| (5) AGA | 4.62 | 1.15 | 2.51 | 4.26 | 2.85 |
| URBAN | 32.50 | 27.02 | 26.54 | 48.93 | 30.41 |
| NON–URBAN | 67.50 | 72.98 | 73.46 | 51.07 | 69.59 |
| TOTAL | 100.00 | 100.00 | 100.00 | 100.00 | 100.00 |
| % OF PROVINCIAL AREA | 3.42% | 1.83 % | 0.54 % | 8.60 % | 0.77 % |

(4) Mineral extraction and dump sites is included in (3) ICT

**Figure 4.** Forms of spatial occupation generated by the urban process. Provinces of Girona, Barcelona and Tarragona.

According to the data shown, the continuous urban fabric (Class (1) CUF) is nourished by 63.60% of arable land, permanent crops, and heterogeneous agricultural areas (Class (6) AAZ). We have previously commented that the pressure on agricultural uses due to the urban process is very great near urban areas, so that the expectations generated by this process, in terms of land prices, prevent crops from resisting it and, therefore, a continued decrease in the profitability of production leads to the loss of the activity and the sale of land for the urban process. The same occurs with the discontinuous urban fabric (Class (2) DUF) and commercial, industrial, and infrastructure land (Class (3) ICT), which obtains its profits from agricultural land (Class (6) AAZ) in proportions of 59.87% and 73.65%, respectively. Another fact to be highlighted is that the continuous urban fabric (Class (1) CUF) shows losses in favour of the discontinuous urban fabric (Class (2) DUF) by about 61.57%. This is due to another fact that we have also mentioned above, which is the generation of large urban land pools that occurred with the urban planning of the 1990s. Finally, much of this programmed land did not become consolidated as a continuous urban fabric but did so in a dispersed manner and disconnected from the continuous urban fabric as a discontinuous urban fabric.

Another reality that can be observed in Table 5 is the close relationship that exists between areas of natural vegetation (forests) and areas of shrub and/or herbaceous vegetation (Class (7) NVA), croplands, permanent crops, and heterogeneous agricultural areas (Class (6) AAZ), and open spaces with little or no vegetation (Class (8) OSV). Their figures for both losses and gains have been marked in bold. These data have a clear meaning: as agricultural activity loses soils due to the pressure of the urban process, it seeks new soils for cultivation in areas with shrub and/or herbaceous vegetation, areas that are not pro-

tected by legislation. At the same time, the pressure suffered by the latter is a determining factor in the processes of change that generate losses and gains in open spaces with little or no vegetation. Thus, 70.97% of the gains of Class (6) AAZ are realized by Class (7) NVA, but paradoxically, 57.77% of the gains of the latter are realized by the former. In this set of relationships, Class (8) OSV evolves in losses and gains, in clear dependence of the two previous ones; however, 83.30% of the gains of this class are produced by natural vegetation zones (forests) and areas of shrub and/or herbaceous vegetation. And 81.30% of the losses in this class are also a function of it.

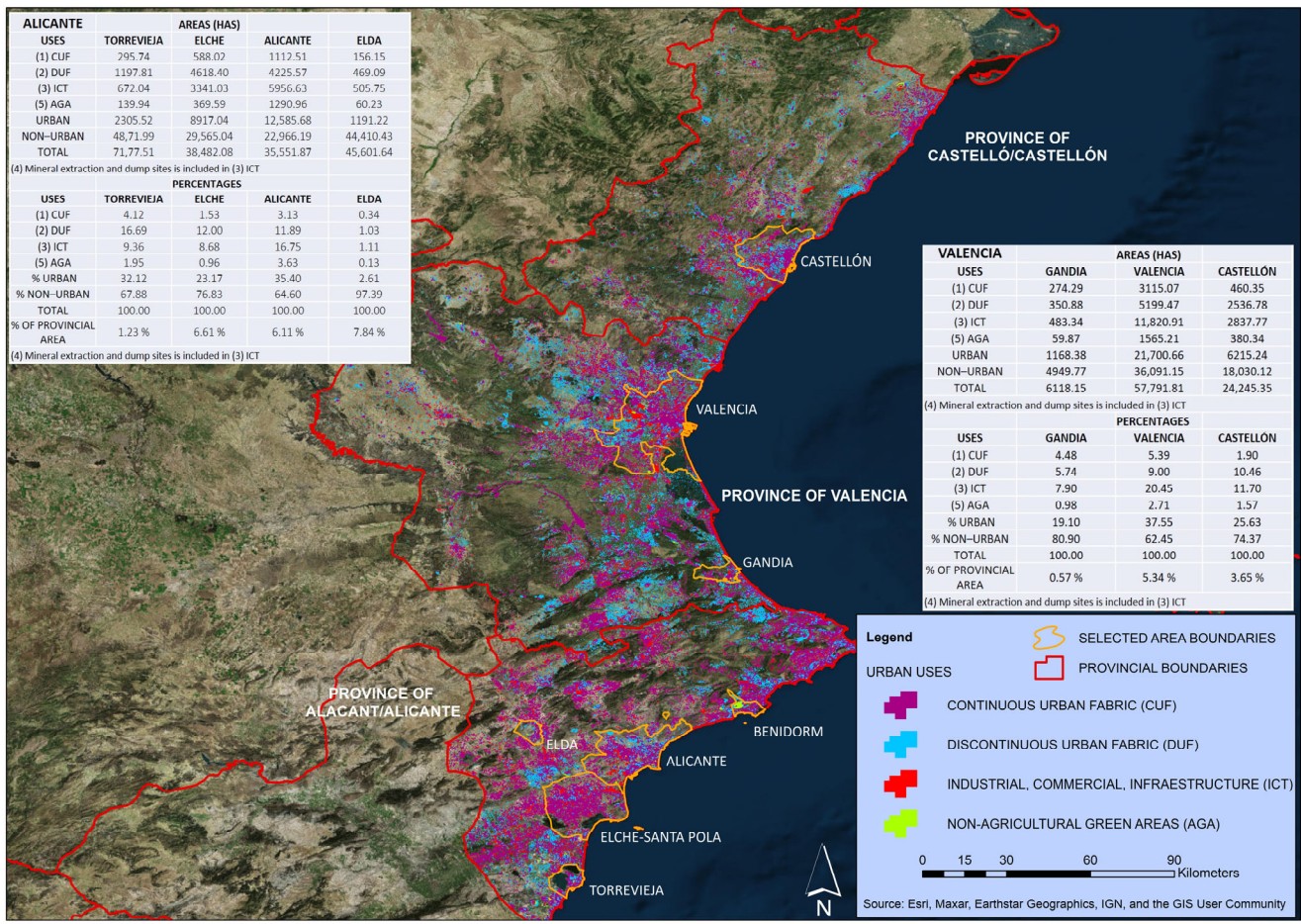

**Figure 5.** Forms of spatial occupation generated by the urban process. Provinces of Castellón, Valencia and Alicante.

In conclusion, the greater or lesser importance of these figures depends on the location of these uses in the territory, since their evolution will depend on their proximity to large urban centres, which will mean that they will be affected by the pressure of the urban process, albeit with different intensities, but in two ways: first, their pressure on agricultural activity generates the abandonment of this activity and, consequently, cultivated soils become spaces with little or no vegetation. Or, if the urban process takes a long time to occupy them, they tend to regenerate naturally and return, to a certain extent, to their original state. Second, the space studied has suffered, during recent years, an intensive occupation process due to the phenomenon of diffuse urbanism, and because of this, buildings or houses have been built in spaces far away from the continuous urban fabric. We have tried to reflect this situation in our work by subdividing the discontinuous urban fabric into different degrees of intensity of occupation, as we will see below.

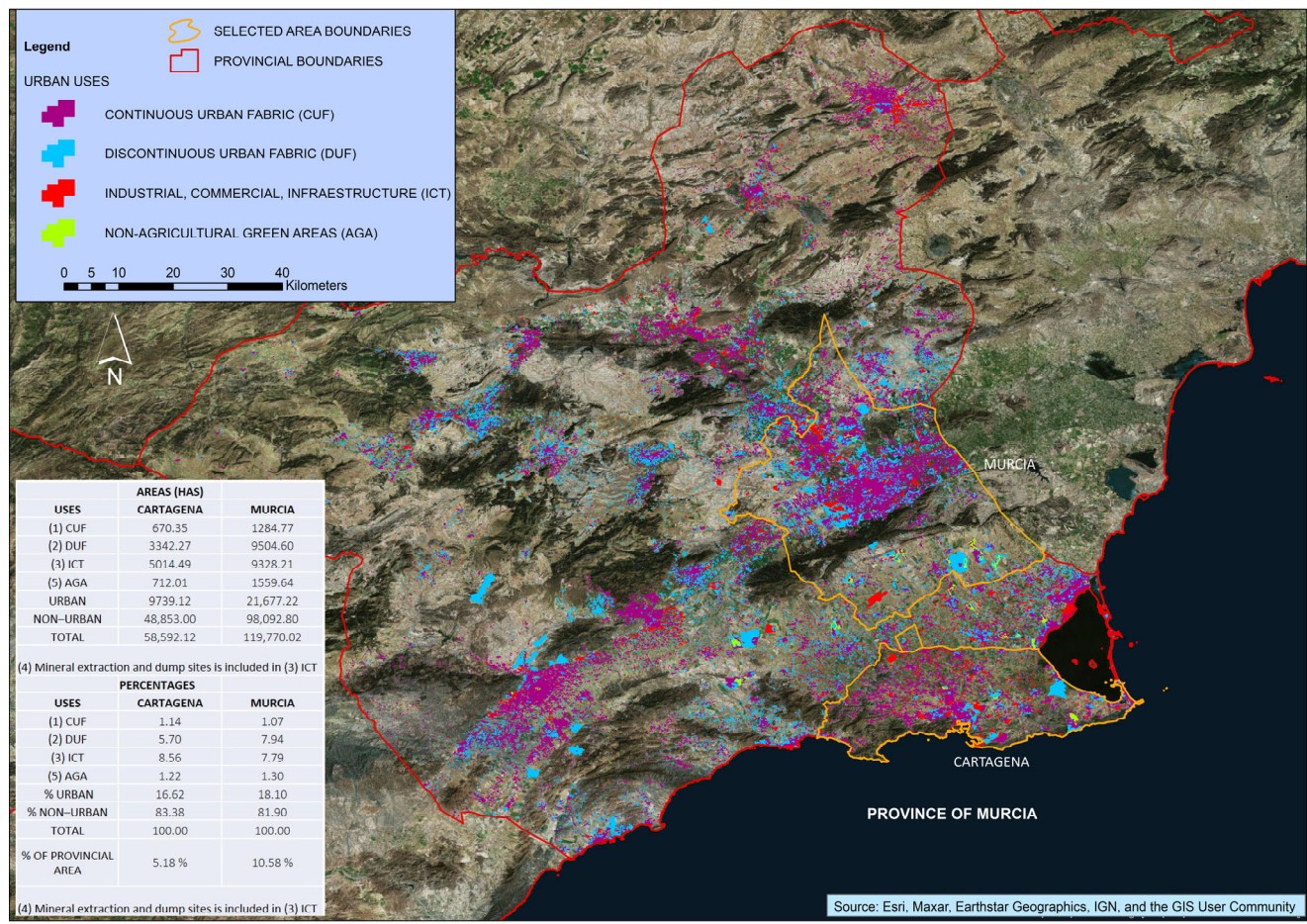

**Figure 6.** Forms of spatial occupation generated by the urban process. Province of Murcia.

**Table 5.** Matrix of types of land use change between 1990 and 2018: areas (has) and percentages of earnings and losses. On the diagonal are marked the areas of unchanged use classes, called persistences; losses are accounted for in the rows according to each use and the columns account for the earnings from each land use.

| AREAS | (1) CUF | (2) DUF | (3) ICT | (4) MAL | (5) AGA | (6) AAZ | (7) NVA | (8) OSV | (9) WWS | LOSSES | TOTAL 1 (A) |
|---|---|---|---|---|---|---|---|---|---|---|---|
| | | | | | SURFACES (HAS) | | | | | | |
| (1) CUF | 51,670 | 24,960 | 7031 | 63 | 1144 | 5311 | 1739 | 134 | 157 | 40,539 | 92,209 |
| (2) DUF | 6690 | 73,696 | 2903 | 313 | 1820 | 11,657 | 15,370 | 239 | 296 | 39,288 | 112,984 |
| (3) ICT | 1563 | 2073 | 25,547 | 89 | 437 | 2583 | 865 | 65 | 128 | 7805 | 33,351 |
| (4) MAL | 37 | 199 | 263 | 6379 | 30 | 659 | 2193 | 44 | 138 | 3564 | 9943 |
| (5) AGA | 140 | 441 | 326 | 0 | 3014 | 157 | 214 | 23 | 58 | 1360 | 4374 |
| (6) AAZ | 20,944 | 78,439 | 59,435 | 7394 | 10,818 | 3,315,682 | 597,027 | 15,859 | 7339 | 797,255 | 4112,937 |
| (7) NVA | 2451 | 20,853 | 6391 | 7061 | 5933 | 260,764 | 3,389,265 | 85,795 | 5590 | 394,838 | 3,784,102 |
| (8) OSV | 880 | 3274 | 2421 | 2016 | 1124 | 83,520 | 412,583 | 128,799 | 1690 | 507,508 | 636,307 |
| (9) WWS | 226 | 782 | 1924 | 13 | 295 | 2759 | 3399 | 832 | 35,370 | 10,230 | 45,600 |
| EARNINGS | 32,931 | 131,021 | 80,694 | 16,949 | 21,601 | 367,410 | 1,033,390 | 102,992 | 15,397 | 1,802,385 | ↓ |
| TOTAL 2 (B) | 84,601 | 204,717 | 106,241 | 23,328 | 24,615 | 3,683,092 | 4,422,655 | 231,792 | 50,767 | → | 8,831,807 |

**Table 5.** *Cont.*

| AREAS | (1) CUF | (2) DUF | (3) ICT | (4) MAL | (5) AGA | (6) AAZ | (7) NVA | (8) OSV | (9) WWS | LOSSES | TOTAL 1 *(A)* |
|---|---|---|---|---|---|---|---|---|---|---|---|
| **SURFACES (HAS)** | | | | | | | | | | | |
| **PERCENTAGE OF EARNINGS** | | | | | | | | | | | |
| EARNINGS(%) | (1) CUF | (2) DUF | (3) ICT | (4) MAL | (5) AGA | (6) AAZ | (7) NVA | (8) OSV | (9) WWS | EARNINGS | % AREA*(E)* |
| (1) CUF | 61.07*(C)* | 19.05 | 8.71 | 0.37 | 5.29 | 1.45 | 0.17 | 0.13 | 1.02 | 2.25 | 1.04 |
| (2) DUF | 20.31 *(D)* | 36.00 | 3.60 | 1.85 | 8.43 | 3.17 | 1.49 | 0.23 | 1.92 | 2.18 | 1.28 |
| (3) ICT | 4.75 | 1.58 | 24.05 | 0.52 | 2.03 | 0.70 | 0.08 | 0.06 | 0.83 | 0.43 | 0.38 |
| (4) MAL | 0.11 | 0.15 | 0.33 | 27.34 | 0.14 | 0.18 | 0.21 | 0.04 | 0.90 | 0.20 | 0.11 |
| (5) AGA | 0.42 | 0.34 | 0.40 | 0.00 | 12.25 | 0.04 | 0.02 | 0.02 | 0.38 | 0.08 | 0.05 |
| (6) AAZ | **63.60** | **59.87** | **73.65** | **43.62** | **50.08** | 90.02 | **57.77** | 15.40 | **47.67** | 44.23 | **46.57** |
| (7) NVA | 7.44 | 15.92 | 7.92 | **41.66** | 27.47 | **70.97** | 76.63 | **83.30** | **36.30** | 21.91 | **42.85** |
| (8) OSV | 2.67 | 2.50 | 3.00 | 11.89 | 5.20 | 22.73 | **39.93** | 55.57 | 10.98 | 28.16 | 7.20 |
| (9) WWS | 0.69 | 0.60 | 2.38 | 0.08 | 1.36 | 0.75 | 0.33 | 0.81 | 69.67 | 0.57 | 0.52 |
|  | 100 | 100 | 100 | 100 | 100 | 100 | 100 | 100 | 100 | 100 | 100 |
| **PERCENTAGE OF LOSSES** | | | | | | | | | | | |
| LOSSES(%) | (1) CUF | (2) DUF | (3) ICT | (4) MAL | (5) AGA | (6) AAZ | (7) NVA | (8) OSV | (9) WWS | LOSSES | % AREA *(F)* |
| (1) CUF | 56.04 | 61.57 | 17.34 | 0.16 | 2.82 | 13.10 | 4.29 | 0.33 | 0.39 | 100 | **43.96** |
| (2) DUF | 17.03 | 65.23 | 7.39 | 0.80 | 4.63 | 29.67 | 39.12 | 0.61 | 0.75 | 100 | 34.77 |
| (3) ICT | 20.03 | 26.57 | 76.60 | 1.14 | 5.61 | 33.10 | 11.08 | 0.84 | 1.65 | 100 | 23.40 |
| (4) MAL | 1.05 | 5.59 | 7.39 | 64.16 | 0.83 | 18.49 | **61.53** | 1.24 | 3.88 | 100 | 35.84 |
| (5) AGA | 10.29 | 32.42 | 23.95 | 0.00 | 68.91 | 11.58 | 15.75 | 1.71 | 4.30 | 100 | 31.09 |
| (6) AAZ | 2.63 | 9.84 | 7.46 | 0.93 | 1.36 | 80.62 | **74.89** | 1.99 | 0.92 | 100 | 19.38 |
| (7) NVA | 0.62 | 5.28 | 1.62 | 1.79 | 1.50 | **66.04** | 89.57 | **21.73** | 1.42 | 100 | 10.43 |
| (8) OSV | 0.17 | 0.65 | 0.48 | 0.40 | 0.22 | 16.46 | **81.30** | 20.24 | 0.33 | 100 | **79.76** |
| (9) WWS | 2.21 | 7.64 | 18.81 | 0.13 | 2.88 | 26.97 | 33.23 | 8.14 | 77.57 | 100 | 22.43 |
| **% LOSSES** | 1.83 | 7.27 | 4.48 | 0.94 | 1.20 | 20.38 | 57.33 | 5.71 | 0.85 | 100 | 20.41 |

A. TOTAL 1 = LOSSES + PERSISTENCES. TOTAL 2 = EARNINGS + PERSISTENCES. B. The proportion of persistence is calculated with respect to TOTAL 2. C. The calculation of the earnings is completed with respect to the total of the EARNINGS. D. This percentage is made with respect to the total surface area (TOTAL 1), so that we can see the weight that each type of land use has in the whole. E. This percentage is made with respect to the total surface area (TOTAL 1), so that we can see the weight of each use in the dynamics of changes. Legend: (1) CUF: continuous urban fabric, (2) DUF: discontinuous urban fabric and areas under construction, (3) ICT: industrial, commercial, and transport zones, (4) MAL: mining areas and landfills, (5) AGA: artificial, non-agricultural green areas, (6) AAZ: farmland, permanent crops, and heterogeneous agricultural areas, (7) NVA: areas of natural vegetation (forests) and areas of shrub and/or herbaceous vegetation, (8) OSV: open spaces with scarce or no vegetation, (9) WWS: continental or coastal wetlands and water surfaces. Data source: Corine Land Cover (CLC). Map of land occupation in Spain, scale 1:100,000, corresponding to the European project Corine Land Cover, versions from 1990 and 2018. National Geographic Institute of Spain. Government of Spain.

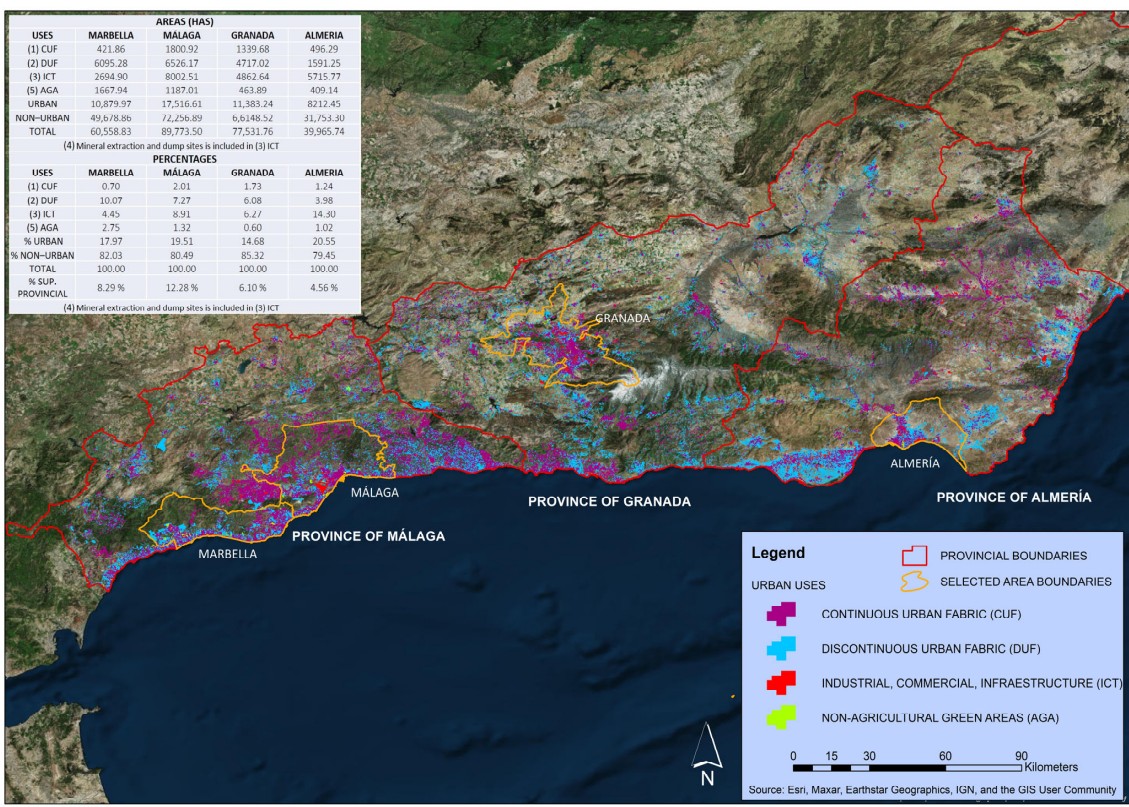

**Figure 7.** Forms of spatial occupation generated by the urban process. Provinces of Almería, Granada and Málaga.

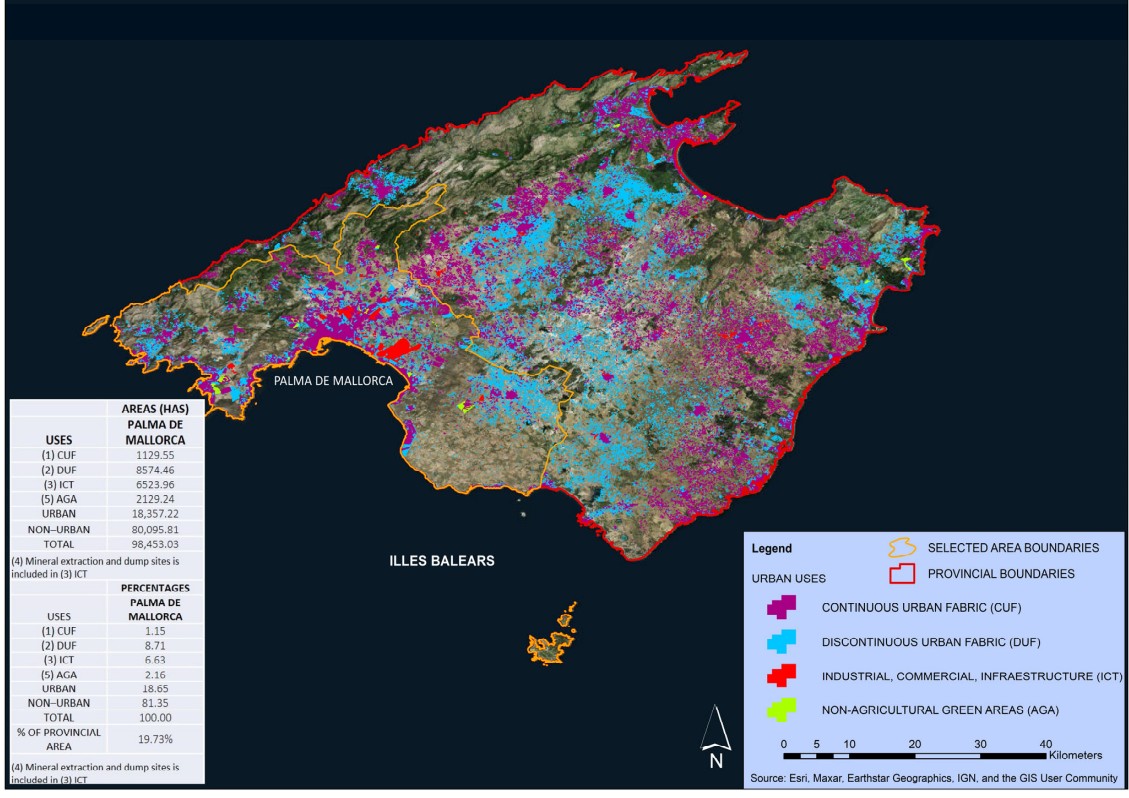

**Figure 8.** Forms of spatial occupation generated by the urban process. Palma de Mallorca (Illes Balears).

*3.2. Analysis of the Spatial Occupation Generated by the Urban Process through the Definition of Units of Urban Functionality and Homogeneous Behaviour*

The territorial development forms that have taken place on the Spanish Mediterranean coast can be specified in a series of forms of occupation that characterize this territory. We can find different magnitudes in these forms of occupation that give rise to units of different morphology, given that they include compact or discontinuous urban spaces, regardless of their size and another series of uses. Table 6 shows their surface areas, quantified according to the different urban structures.

**Table 6.** Surface and typology of URBAN land uses, 2017, for the analysed areas of the study area. Areas in has. Data sorted by surface area of urban uses.

| SPACES/USES | (1) CUF | (2A) DUF | (2B) DUF | (2C) DUF | (2D) DUF | (3) ICT | (4) MAL | (5) AGA | TOTAL URBAN |
|---|---|---|---|---|---|---|---|---|---|
| **Traditional urban centres and their successive extensions** | | | | | | | | | |
| **Barcelona** | 8756.32 | 7731.60 | 7670.04 | 7143.90 | 4783.00 | 33,640.35 | 2166.84 | 6029.69 | 77,921.74 |
| **Valencia** | 3482.43 | 1552.54 | 1467.19 | 3279.51 | 3392.04 | 15,064.14 | 763.49 | 2038.91 | 31,040.25 |
| **Palma** | 1355.37 | 2170.15 | 2551.40 | 3442.70 | 6614.53 | 9208.02 | 556.88 | 2355.37 | 28,254.42 |
| **Murcia** | 1284.97 | 1602.44 | 1309.72 | 2273.73 | 4323.93 | 8564.67 | 774.73 | 1559.79 | 21,693.98 |
| **Málaga** | 1978.46 | 1738.14 | 1643.79 | 1508.36 | 3012.40 | 8241.52 | 826.21 | 1219.40 | 20,168.28 |
| **Granada** | 1427.37 | 1745.71 | 930.78 | 1298.44 | 1956.19 | 5427.16 | 649.02 | 484.25 | 13,918.92 |
| **Alicante** | 1112.81 | 1399.03 | 1442.81 | 665.81 | 717.97 | 5104.31 | 854.47 | 1320.01 | 12,617.22 |
| **Almería** | 496.73 | 337.17 | 211.42 | 112.45 | 931.97 | 5245.38 | 509.13 | 412.51 | 8256.76 |
| **Castellón** | 547.63 | 370.02 | 337.58 | 806.29 | 1470.57 | 3187.20 | 226.68 | 468.26 | 7414.23 |
| **Tarragona** | 446.64 | 563.84 | 376.72 | 521.5 | 621.93 | 3416.70 | 65.6 | 1000.58 | 7013.51 |
| **Girona** | 225.51 | 200.02 | 117.82 | 89.13 | 109.43 | 467.9 | 52 | 131.66 | 1393.47 |
| **Suburban tourist city spaces** | | | | | | | | | |
| **Marbella** | 432.16 | 632.94 | 974.36 | 1711.24 | 3053.65 | 2497.82 | 343.42 | 1677.38 | 11,322.97 |
| **Torrevieja** | 295.7 | 336.58 | 289.02 | 207.43 | 365.78 | 671.13 | 6.44 | 166.43 | 2338.51 |
| **Benidorm** | 146.25 | 219.42 | 74.56 | 92.35 | 181.66 | 610.08 | 4.82 | 241.05 | 1570.19 |
| **Gandía** | 274.16 | 97.75 | 61.21 | 24.39 | 167.48 | 483.49 | 0.71 | 63.53 | 1172.72 |
| **Modern and complex suburban spaces** | | | | | | | | | |
| **Cartagena** | 680.64 | 756.19 | 523.05 | 580.53 | 1734.37 | 4646.14 | 427.07 | 773.69 | 10,121.68 |
| **Elche–Sta. Pola** | 588.25 | 299.3 | 254.02 | 621.92 | 3446.72 | 3236.86 | 106.16 | 369.51 | 8922.74 |
| **Reus** | 247.31 | 149.98 | 126.87 | 183.7 | 727.67 | 1503.06 | 59.08 | 132.86 | 3130.53 |
| **Elda** | 160.71 | 26.99 | 29.44 | 184.83 | 231.26 | 369.95 | 145.83 | 60.22 | 1209.23 |
| **Manresa** | 199.74 | 76.46 | 15.64 | 14.09 | 113.19 | 548.59 | 27.78 | 103.94 | 1099.43 |
| **TOTAL** | 24,139.15 | 22,006.27 | 20,407.44 | 24,762.31 | 37,955.73 | 112,134.45 | 8566.34 | 20,609.02 | 270,580.71 |

Legend: (1) CUF: continuous urban fabric (>80%), (2A) DUF: discontinuous dense urban fabric (50–80%), (2B) DUF: discontinuous medium density urban fabric (30–50%), (2C) DUF: discontinuous low density urban fabric (10–30%), (2D) DUF: discontinuous very low density urban fabric (<10%), isolated structures, construction sites, and land without current use, (3) ICT: industrial, commercial, and transport zones (industrial, commercial, public, military, private units, transit roads and associated land, port areas, and airports), (4) MAL: mining, landfill, and construction areas, (5) AGA: artificial, non-agricultural green areas. Data source: land occupation map of the largest Spanish urban areas at a scale of 1:15,000, corresponding to the Territory Service of the European Copernicus Programme, versions 2006 and 2012 (URBAN COVER). Updated to High Resolution Information System on Land Occupation of Spain (SIOSE HR), integrated within the National Plan for Territorial Observation (PNOT), 2017. National Geographic Institute of Spain. Government of Spain.

The magnitudes of the non-urban spaces in the different territories of the study area acquire great importance, since these spaces are considered as reserve soils that will support

the urban process in due course. To this end, we include Table 7, since two reflections can be drawn from it: one, that the future dynamics of urban development will depend on the availability of spaces for its development, spaces that will be supplied fundamentally by the spaces currently occupied by agricultural activity (column (6) AAZ), depending on the capacity of agricultural activity to withstand urban pressure. And two, that the persistence of natural spaces (column (7) NVA) will be crucial as spaces that will serve as containment to the trend of occupation of the whole territory, characteristic of the urban process, providing, in turn, quality of life environments to the urban fabric in a context of climate change so accentuated in Mediterranean areas. On the other hand, it will be interesting to observe the evolution of open spaces (column (8) OSV) in the future, since the assessment of the environmental deterioration of the natural spaces near these continuous urban fabrics will depend to a large extent on this. Another element that can lead to a very positive assessment of the occupation of the territory will be the evolution of wetlands and water areas (column (9) WWS), since, in an environment that tends to aridity and desertification, the conservation of these spaces will be crucial to prevent their disappearance.

**Table 7.** Surface and typology of NON–URBAN land uses, 2017, for the analysed areas of the study area. Areas in has. Data sorted by surface area of non–urban uses.

| SPACES/USES | (6) AAZ | (7) NVA | (8) OSV | (9) WWS | TOTAL, NON URBAN | TOTAL, AREAS (*) |
|---|---|---|---|---|---|---|
| Traditional urban centres and their successive extensions | | | | | | |
| **Palma** | 141,036.24 | 28,635.00 | 1814.66 | 530.56 | 172,016.46 | 200,270.86 |
| **Barcelona** | 80,053.57 | 84,667.65 | 450.12 | 958.16 | 166,129.50 | 244,051.23 |
| **Granada** | 116,062.60 | 19,209.89 | 6.38 | 511.99 | 135,790.86 | 149,709.77 |
| **Málaga** | 121,250.29 | 9255.74 | 571.42 | 939.5 | 132,016.95 | 152,185.24 |
| **Murcia** | 90,844.33 | 6776.10 | 0 | 571.44 | 98,191.87 | 119,885.86 |
| **Valencia** | 67,732.14 | 2551.93 | 188.38 | 3040.30 | 73,512.75 | 104,552.99 |
| **Almería** | 29,963.15 | 523.81 | 382.34 | 956.96 | 31,826.26 | 40,083.01 |
| **Castellón** | 20,729.80 | 3945.29 | 77.05 | 59.56 | 24,811.70 | 32,225.93 |
| **Alicante** | 22,417.01 | 189.29 | 230.69 | 127.73 | 22,964.72 | 35,581.94 |
| **Tarragona** | 10,657.83 | 3329.66 | 109.11 | 60.37 | 14,156.97 | 21,170.47 |
| **Girona** | 937.16 | 1515.23 | 0 | 41.98 | 2494.37 | 3887.83 |
| Suburban tourist city spaces | | | | | | |
| **Marbella** | 32,354.37 | 24,358.64 | 369.12 | 285.45 | 57,367.58 | 68,690.56 |
| **Gandía** | 4172.29 | 706.4 | 37.42 | 11.91 | 4928.02 | 6100.73 |
| **Torrevieja** | 2128.43 | 17.28 | 404.27 | 2287.90 | 4837.88 | 7176.37 |
| **Benidorm** | 1380.51 | 816.28 | 67.12 | 18.63 | 2282.54 | 3852.73 |
| Modern and complex suburban spaces | | | | | | |
| **Cartagena** | 45,472.75 | 2945.29 | 172.73 | 408.98 | 48,999.75 | 59,121.42 |
| **Elche–Sta. Pola** | 27,637.71 | 28.7 | 31.09 | 1851.68 | 29,549.18 | 38,471.93 |
| **Reus** | 7387.77 | 946.3 | 79.2 | 0 | 8413.27 | 11,543.81 |
| **Elda** | 3207.38 | 163.65 | 0 | 1.74 | 3372.77 | 4581.99 |
| **Manresa** | 2200.65 | 838.33 | 0 | 26.74 | 3065.72 | 4165.14 |
| **TOTAL** | 827,625.98 | 191,420.45 | 4991.11 | 12,691.58 | 1,036,729.12 | 1,307,309.85 |

(*) TOTAL AREAS is the sum of TOTAL, URBAN (Table 6) and TOTAL, NON–URBAN (Table 7). Legend: (6) AAZ: agricultural areas, farmland, permanent crops, and heterogeneous agricultural areas, (7) NVA: areas with natural vegetation (areas of natural vegetation (forests) or areas of shrub and/or herbaceous vegetation), (8) OSV: open spaces with scarce or no vegetation, (9) WWS: continental or coastal wetlands and water surfaces. Data source: land occupation map of the largest Spanish urban areas at a scale of 1:15,000, corresponding to the Territory Service of the European Copernicus Programme, versions 2006 and 2012 (URBAN COVER). Updated to High Resolution Information System on Land Occupation of Spain (SIOSE HR) integrated within the National Plan for Territorial Observation (PNOT), 2017. National Geographic Institute of Spain. Government of Spain.

For an analysis of the territorial distribution of uses, Figures 4–8 below can be seen. In them, we can observe the forms of the process whose magnitudes have been quantified in Tables 6 and 7 above. On the one hand, there are a series of large traditional urban centres and their successive extensions, which have been classified in class (1) CUF: continuous urban fabric (>80%), which, ordered by surface area, are Barcelona, Valencia, Palma, Murcia, Malaga, Granada, Alicante, Almeria, Castellon, Tarragona, and Girona. They are characterized by being urban areas that assume central functions and concentrate urban and administrative functionality. They are the original nucleus from which later urban developments have taken place, in the form of urban extensions. From these large urban areas, forms of land occupation have been generated that could be called first peripheries and have been quantified fundamentally in class (2A) DUF: discontinuous dense urban fabric (50–80%). These are spaces corresponding to the growth of the 1980s and 1990s; they are models of open and high-rise buildings, but which are forming continuous and compact urban fabrics. These urban spaces acquire a similar functionality to the traditional urban centres and form part of the structure of the consolidated urban space.

The extension of the urban process beyond the traditional nucleus and their first peripheries gives rise to new discontinuous urban spaces and gives rise to what have been called residential units in formation. They are constituted by the processes of unconnected residential growth of the urban fabric and that rely on the first sections of the main road axes for their organization. Their surface areas have been quantified in classes (2B) DUF: discontinuous medium density urban fabric (30–50%), (2C) DUF: discontinuous low density urban fabric (10–30%), and (3) ICT: industrial, commercial, and transport zones (industrial, commercial, public, military, private units, transit roads and associated land, port areas, and airports). They can be located from the first peripheries or be territorially isolated from the urban fabric and therefore do not present territorial continuity. They are urban spaces that arise by occupying vacant space or replacing existing uses, mainly agricultural. They usually extend between neighbouring municipalities from the large urban centres, giving rise to a model of urban expansion of metropolitan order. In these spaces are located the industrial peripheries, which are units formed by the consolidated agglomeration of industrial estates or land for industrial use, generally compact. The last ones will be located mainly on the urban edge and near major road infrastructures.

Finally, the urban process is extended in rural or agricultural areas by means of isolated dwellings or housing nuclei in self-construction processes on agricultural plots, which we have included in class (2D) DUF: discontinuous very low density urban fabric (<10%), isolated structures, construction sites, and land without current use. Here are also located other non-residential constructions such as warehouses or buildings related to agricultural or industrial activity, which produce considerable territorial deterioration if they are executed out of planning or without the corresponding urban services.

As can be seen in Figures 4–8 below, the occupation of the territory is carried out in an unconnected manner, which denotes a lack of planning. The modification of the territory through the introduction of the city's own forms is generated by a gathering of real estate properties destined to different uses and served by various networks. These networks multiply as the central urban space, the so-called large traditional urban centres and their successive extensions, has increased in size and has become more complex. We can then observe how the urban space is too extensive and cannot be perceived as unique but must be seen as a system of diverse spaces. But not only does the urban space break the ecological balance, but also the deterioration of agricultural spaces included in the class (6) AAZ: agricultural surfaces, arable land, permanent crops, and heterogeneous agricultural areas when the activity has not married the soil, but exploits it, becoming a source of economic production, which probably in the near future will not be able to compete with urban land, which will lead to its transformation and loss of functionality.

The extension of the urban process from the aforementioned centres by the impulse of real estate activity has given rise to units of consolidated conurbation, known as suburban tourist areas. These are continuous or discontinuous urban spaces in transformed spaces

that give rise to an urban or tourist continuum that is closer, in forms and functions, to a traditional city morphology than to a tourist space, strictly speaking. In their development, they give rise to a characteristic morphology, such as the suburban tourist city. They are usually formed from an agglomeration of urbanizations that have been emerging independently and are dispersed extensively as autonomous fabrics of tourist accommodation (mainly tourist homes or second homes). In many cases, they present a rudimentary territorial structure, excessively supported by main roads, with very limited accessibility in many cases, without adequate facilities, without central places or complementary offerings, outside the traditional nucleus, often far from them, with important expansive tensions and a strong tendency towards urban densification when urban renewal processes are developed, through an increase in the buildability of buildings. In this case, the vacant spaces between consolidated or consolidating urban fabrics are noteworthy, due to the importance of their strategic location. These are spaces in which the previous uses are maintained, but with a marked tendency to disappear and with important pressures of urban occupation. They form a compact and mixed nucleus of permanent residences, tourist lodgings, and services with an integrated urban function, with respect to the nucleus or zones of defined and recognizable centrality. Marbella, Torrevieja, Benidorm, and Gandía have been included in this type of morphology.

Other forms of occupation that can be observed are modern and complex suburban spaces. These are units that give shape to spaces characterized by heterogeneous processes of implantation of industrial activities in low-density fabrics: isolated or semi-detached single-family housing, agricultural and livestock facilities, industrial parks, etc., on a plot structure of rustic origin, forming structures of relatively autonomous operation, with a tendency to form an agglomerate on a rudimentary territorial structure based on traditional rural roads that interconnect with secondary road axes. They intermingle rustic forms and elements of clearly urban morphology but with deficient urban services, as a consequence, in many cases, of their origin on a nucleus or traditional population entity or a rustic or irregular parcelling. They are distant or individualized from the continuous urban spaces and framed in the urban peripheries and nearby rural spaces. Here we can cite a series of nuclei that have developed as a result of industrial activity, such as Elche, Reus, Elda, and Manresa.

The proximity of agrarian spaces to areas of very dynamic urban expansion generates a typology that we could call spaces of urban vocation with the presence of rural uses. These are areas with a purely urban vocation based on the abandonment of rural activity, where the presence of the urban process can be seen through the implantation of housing for residential use in scattered areas, without any link to agricultural activity, supported by the network of rural roads, and therefore lacking an urban design of their own.

The support that gives structure to this aggregate of typologies are the infrastructures and other general facilities: commercial, sports, or industrial complexes, which, as mentioned above, ensure the functioning of the whole and relate the parts to each other. The construction of the territorial model cannot ignore these elements and must take into account both the identification of each one of them and the value and function it fulfils, as well as its capacity to resolve the relationship and articulation between the parts of the territory. Under such criteria, we must identify the urban axes that support mobility and urban activity and the road systems or elements of the territorial structure.

### 3.3. Evaluation of the Degree of Spatial Adequacy of the Urban Process Based on the Proposed Indicators

In this section, we explain the results obtained from the application of the multi-criteria evaluation process discussed in Section 2.3.3 of the methodology. As mentioned above, a series of indicators were proposed, on the basis of which the multi-criteria evaluation process was carried out. Table 8 shows the results. The surface areas were quantified. Table 9 shows the proportions of these surfaces. Figures A1–A3 of Appendix A show the results of the multi-criteria evaluation process in each of the areas studied.

**Table 8.** Assessment of the adequacy of the urban process, according to the categories of the distance from the ideal point. Surfaces for the analysed areas of the study area. Suitability criteria: Very Close, as the best valued, to A Long Way, as the worst rated. Areas in has. Data sorted by surface area (TOTAL). The predominant class has been highlighted in bold.

| AREAS | VERY CLOSE | CLOSE | LITTLE CLOSE | MIDPOINT | NOT FAR | FAR | A LONG WAY | TOTAL |
|---|---|---|---|---|---|---|---|---|
| Traditional urban centres and their successive extensions | | | | | | | | |
| Murcia | **61,235** | 4220 | 8256 | 11,578 | 31,322 | 939 | 2219 | 119,770 |
| Palma M. | 32,297 | **39,407** | 5146 | 2927 | 6151 | 9810 | 2714 | 98,453 |
| Málaga | **64,243** | 6104 | 1318 | 915 | 11,890 | 2356 | 2948 | 89,774 |
| Granada | **23,479** | 4278 | 13,521 | 8808 | 22,461 | 2569 | 2415 | 77,532 |
| Barcelona | **34,607** | 2053 | 11,369 | 3833 | 9834 | 1647 | 3168 | 66,511 |
| Valencia | **15,341** | 7212 | 8179 | 6612 | 15,300 | 2464 | 2684 | 57,792 |
| Almería | 1780 | 12,080 | 136 | 73 | 132 | 3559 | **22,206** | 39,966 |
| Alicante | 7222 | 3108 | 1499 | 4846 | **12,842** | 2975 | 3060 | 35,552 |
| Castellón | 5462 | 3718 | 2962 | **8764** | 1686 | 381 | 1274 | 24,245 |
| Tarragona | 5731 | 1272 | 2014 | 1296 | **9357** | 726 | 1141 | 21,537 |
| Girona | **2255** | 193 | 336 | 476 | 424 | 225 | 651 | 4560 |
| Suburban tourist city spaces | | | | | | | | |
| Marbella | **28,228** | 2217 | 4050 | 1180 | 20,063 | 4168 | 746 | 60,653 |
| Benidorm | **6558** | 5256 | 5611 | 5101 | 5656 | 5214 | 5140 | 38,536 |
| Torrevieja | 144 | **3598** | 1950 | 727 | 542 | 54 | 162 | 7178 |
| Gandía | 468 | 165 | 465 | **4331** | 313 | 78 | 299 | 6118 |
| Modern and complex suburban spaces | | | | | | | | |
| Cartagena | 13,201 | 3307 | 397 | 363 | 169 | **38,570** | 2585 | 58,592 |
| Elda | 18 | 23 | 5919 | 9292 | 6282 | 11,946 | **12,122** | 45,602 |
| Elche-Sta.Pola | 5761 | 4599 | 568 | 4694 | **20,796** | 623 | 1441 | 38,482 |
| Reus | 2764 | 286 | 940 | 1409 | **4315** | 1151 | 705 | 11,571 |
| Manresa | **2044** | 195 | 89 | 398 | 69 | 1263 | 118 | 4176 |
| Total Area | 312,840 | 103,292 | 74,727 | 77,623 | 179,603 | 90,717 | 67,797 | 906,598 |

Data sources: Corine Land Cover (CLC), Geographic Population Reference Information (IGRPP), and Urban Atlas (URBAN COVER) with High Resolution Land Occupation Information System in Spain (SIOSE HR).

Table 9 shows the percentages of the seven categories considered in the multi-criteria evaluation analysis. The interpretation to be made of this table is that the large traditional urban centres with the highest figures in the Very Close and Close categories are characterized by the fact that the areas they cover contain large vacant spaces that are not yet urbanized, where the uses are diluted or dispersed in the territory and where forest or agricultural spaces predominate, corresponding to classes (2B) DUF: discontinuous medium density urban fabric (30–50%) and (2C) DUF: discontinuous low density urban fabric (10–30%), while the values appearing in the Not Far category correspond to class (1) CUF: continuous urban fabric (>80%). The higher or lower percentage of adequacy in this case will depend on the densities of inhabitants and dwellings and on the volume of urban green areas dedicated to their inhabitants, related to class (5) AGA: non-agricultural artificial green areas. In these data, Malaga stands out, with a figure of 71.56 in the Very Close category, because its municipality contains several protected natural areas that have the functionality of peri-urban parks. The cases of Barcelona with 52.03%, Murcia with

51.13%, and Girona with 49.45% of its surface area in this class respond to similar aspects as those mentioned for Malaga; in the first case, the city of Barcelona presents large spaces that function as urban or peri-urban parks, in the case of Murcia, there is a large part of its surface area with vacant spaces, and in the case of Girona there are spaces with a natural vocation next to the continuous urban fabric. On the other hand, there are areas that have undergone more intensive urban development, such as Alicante and Tarragona, which have the highest values in the Not Far category, with 36.12% and 43.45%, respectively.

**Table 9.** Assessment of the adequacy of the urban process according to the categories of the distance from the ideal point. Percentages for the analysed areas of the study area. Suitability criteria: Very Close, as the best valued, to A Long Way, as the worst rated. The predominant class has been highlighted in bold.

| AREAS | VERY CLOSE | CLOSE | LITTLE CLOSE | MIDPOINT | NOT FAR | FAR | A LONG WAY | TOTAL |
|---|---|---|---|---|---|---|---|---|
| **Traditional urban centres and their successive extensions** | | | | | | | | |
| Murcia | **51.13** | 3.52 | 6.89 | 9.67 | 26.15 | 0.78 | 1.85 | 100 |
| Palma M. | 32.80 | **40.03** | 5.23 | 2.97 | 6.25 | 9.96 | 2.76 | 100 |
| Málaga | **71.56** | 6.80 | 1.47 | 1.02 | 13.24 | 2.62 | 3.28 | 100 |
| Granada | **30.28** | 5.52 | 17.44 | 11.36 | 28.97 | 3.31 | 3.12 | 100 |
| Barcelona | **52.03** | 3.09 | 17.09 | 5.76 | 14.79 | 2.48 | 4.76 | 100 |
| Valencia | **26.55** | 12.48 | 14.15 | 11.44 | 26.47 | 4.26 | 4.64 | 100 |
| Almería | 4.45 | 30.23 | 0.34 | 0.18 | 0.33 | 8.90 | **55.56** | 100 |
| Alicante | 20.32 | 8.74 | 4.22 | 13.63 | **36.12** | 8.37 | 8.61 | 100 |
| Castellón | 22.53 | 15.33 | 12.22 | **36.15** | 6.95 | 1.57 | 5.25 | 100 |
| Tarragona | 26.61 | 5.90 | 9.35 | 6.02 | **43.45** | 3.37 | 5.30 | 100 |
| Girona | **49.45** | 4.23 | 7.37 | 10.43 | 9.30 | 4.94 | 14.27 | 100 |
| **Suburban tourist city spaces** | | | | | | | | |
| Marbella | **46.54** | 3.66 | 6.68 | 1.95 | 33.08 | 6.87 | 1.23 | 100 |
| Benidorm | **17.02** | 13.64 | 14.56 | 13.24 | 14.68 | 13.53 | 13.34 | 100 |
| Torrevieja | 2.01 | **50.13** | 27.17 | 10.13 | 7.55 | 0.75 | 2.25 | 100 |
| Gandía | 7.65 | 2.70 | 7.60 | **70.79** | 5.11 | 1.27 | 4.88 | 100 |
| **Modern and complex suburban spaces** | | | | | | | | |
| Cartagena | 22.53 | 5.64 | 0.68 | 0.62 | 0.29 | **65.83** | 4.41 | 100 |
| Elda | 0.04 | 0.05 | 12.98 | 20.38 | 13.78 | 26.20 | **26.58** | 100 |
| Elche-Sta. Pola | 14.97 | 11.95 | 1.48 | 12.20 | **54.04** | 1.62 | 3.74 | 100 |
| Reus | 23.89 | 2.47 | 8.13 | 12.18 | **37.29** | 9.94 | 6.09 | 100 |
| Manresa | **48.95** | 4.67 | 2.13 | 9.52 | 1.66 | 30.24 | 2.82 | 100 |
| **Total Area** | 34.51 | 11.39 | 8.24 | 8.56 | 19.81 | 10.01 | 7.48 | 100 |
| Mean | *28.57* | *11.54* | *8.86* | *12.98* | *18.97* | *10.34* | *8.74* | |
| Std | *19.37* | *13.36* | *6.91* | *15.83* | *15.69* | *15.32* | *12.49* | |

Data sources: Corine Land Cover (CLC), Geographic Population Reference Information (IGRPP), and Urban Atlas (URBAN COVER) with High Resolution Land Occupation Information System in Spain (SIOSE HR).

Regarding the areas considered to be suburban tourist areas, we must consider their genesis; since they are areas that emerged from the aggregation of tourist developments or are tourist centres, they present a certain disparity in belonging to one category or another, distributing their surfaces in all classes in a proportional way. Thus, in the case of Mabella, where 46.54% of its surface area appears in the Very Close category, we must consider that this

is due to its own configuration, insofar as there are spaces in this area that have undergone little transformation towards the interior, having centred a large part of the urban process on the coastal strip, while towards the interior the urban concentration is diluted, alternating spaces that are still natural spaces with vacant spaces. In the case of Benidorm, which concentrates 17.02% of its surface area in this class, we can say that this is due to its own urban configuration as a space that has concentrated its urban fabric around its original nucleus; this is also the case for Torrevieja with 50.13% of suitability in the Close category, since it presents a concentration of urban spaces on the coast and dilutes this presence towards the interior.

Finally, there are the cases of the spaces that we have called modern and complex suburban spaces, which, in the processes of urban and industrial occupation, have gained their own territorial configuration, yielding higher percentages of surfaces with poor adequacy, as can be seen in Table 9. However, here, the case of Manresa stands out, with a figure of 48.95% in the Very Close category, because this area has low population and housing densities, as can be seen in Tables A1 and A2 of Appendix B.

As has been observed, the configuration of each of these areas and their adaptation to the territory is a function of the ways in which they have designed and developed their urban planning over the last 50 years, since, in some cases, solutions have been adopted that have been more successful for urban development, in relation to the physical environment, whereas in others, this appears crumbled and dispersed. The basis of these statements is what has been observed at the time when the uses were quantified, classified, and categorized into areas, when the changes were evaluated, and when the population and housing densities were fixed. In many circumstances, it may seem that the growth of cities and urban spaces is spontaneous, when, in fact, it is consciously planned. At present, the different uses seem incompatible, and urban and agricultural spaces should be considered in planning as dependent rather than separate elements, when they are related to each other and to the number of inhabitants and dwellings, as the analysis carried out and discussed above has shown.

As previously mentioned, a series of indicators were proposed, based on which multi-criteria evaluation process was applied and which corroborated what was stated. These indicators corroborate what was stated. Tables A1–A3 of Appendix B show their maximum values grouped in the categories that have been established.

## 4. Discussion

The results we obtained have made it possible to define the morphology of the territory through units of homogeneous functionality and to assess their suitability to the territory, which gives foundation to the notion of system, and, at the same time, makes it possible to observe the territorial articulation. With this information, it is possible to define the new relationships between the parts, based on what has been observed in the territory, and to predict the trend of future urban development.

In addition, with the proposed methodology, it is possible to offer results that show the structure that shapes the aggregate of urban uses, which ensures its functioning in the support offered by the infrastructures and other general elements, such as the large facilities, which ensure the functioning of the whole and relate the parts to each other. Thus, an orderly vision is offered, which will not only explain the functioning of each area according to its classification, but also expose the most characteristic value of each one, according to its suitability to the territory. Complementarily, it can be elucidated if it is possible to assume other uses, complementary or equally principal, according to the evaluation of the adequacy, or in any case, to design how new implantations should be made to avoid saturations in the zones of greater urban occupation.

The magnitude of urban phenomena has led to the widespread use of methodologies to measure modern urban forms, with the number of indicators used by different authors for this purpose being variable. Such is the case of Abrantes et al. [1], who proposed five spatial indicators related to density, proximity, and variation in forms during the period 1990–2006, in a methodology remarkably like the one we have used. Along the same lines

is the work of Schwarz [58], whose methodological aim was to identify a minimum set of indicators to analyse the urban form of European cities. To this end, a factor analysis was carried out, based on six indicators for their classification by size, density, and clustering level. As in our work, land cover data from Corine Land Cover were combined with population data. The spatial structure of an urban agglomeration is an abstract expression of the relationship of its urban process, as Xie et al. [9] argue, and determining the effect of the urban process on the territory is of great interest, so it is crucial to provide information on how cities and their surrounding spaces grow. This has led to a growing interest in modelling urban growth using various techniques, as proposed by Makse et al. [30]. The integration of spatial metrics can help to examine and quantify the structural dimensions of land use changes and urban structure, as Liu and Yang [59] do.

Several techniques have been used in this work with the intention already expressed. However, we were interested in automatically obtaining spatial relationships to analyse the territory. Particularly noteworthy for their usefulness are the adya-science or neighbourhood relations between types of urban spaces. We applied the methodologies of Xia et al. [60], Quijada-Alarcón et al. [61], and Attwell and Fletcher [62] to figure out the forms of spatial occupation generated by the urban process in the territorial analysis, and, thanks to them, we were able to define units of urban functionality and homogeneous behaviour.

To support this technique and to verify that the results were as expected, we have calculated the map similarity. The explanations made in various sections of this text would not have been possible without the application of this other technique, as it was fundamental for us to decide the similarity between the different maps. While a visual and subjective analysis can identify general relationships, we needed a quantitative cartographic approach to carry out a detailed and rigorous analysis that would allow us to extract all the information represented between the different maps included in this work. These techniques have long been used by researchers with reliable results, as shown in the work of Cook et al. [63], Dress et al. [64], and Cui et al. [65].

Finally, the application of the multi-criteria evaluation process is a contribution to this type of analysis, as it takes on the complicated challenge of evaluating, by quantification, the implications that the urban forms of the urban process have on the territory. We consider that the actions carried out have effects on the quality of life of their inhabitants, in addition to other environmental effects, and it is therefore necessary, in addition to carrying out a rigorous analysis, to apply techniques that help this analysis; in our case, statistics or mathematics based on GIS and multi-criteria evaluation. As we showed in Table 1, the variability of the distribution of their inhabitants and their demographic characteristics could be factors or elements to consider in future studies. Specifically, the most widespread multi-criteria evaluation techniques were applied, such as the analytical hierarchical process (AHP), which can be ascribed to the paradigm of multi-criteria procedural rationality. It has been applied by authors such as Mendoza et al. [66] and Vaidya and Kumar [67].

However, we must recognise that doubts arise when designing methodologies of this type for practical use: how to make a scientific approach for such complex aspects as those dealt with in this work or what considerations are best when proving scores or weights, as studied by Krejčí and Stoklasa [68]. For example, recourse was made to mathematical methods which, as is well known, involve abstract simplifications of a reality, in this case geographical space (which is understood to be complex), in which only part of it is incorporated. To re-solve this question, a series of techniques were applied that are valid exclusively under the assumptions in which the mathematical or statistical modelling has been proposed [68,69].

However, many models could serve as a basis, and, in this sense, matrix methods were used for data structuring and multi-criteria methods for data analysis. While it is true that there is a wide variety of techniques, it is also true that data management was carried out following the approaches proposed by optimisation methods, including here the theory of optimal suitability or optimal allocation of land uses. In our case, when using multi-criteria techniques, we started from the assumption that a set of approaches was

available, in which the scientific perspective should not be lost when evaluating structured concrete situations. We set out to analyse a complex and unstructured real problem, which needed a clear answer. To do this, it is obviously not possible to retouch reality to apply analytical techniques, such as those used in this work. Therefore, it must be understood that we do not contemplate the existence of a single answer, the one we have given, but that there may be many others that will be associated with the perceptions of reality of other researchers. Therefore, the application of the methodology has sought to combine the goal with the subjective, in such a way as to achieve an objective treatment of the subjective, and thus a rational treatment of a reality that is highly dependent on human behaviour, and therefore close to the intangible. As Moreno [70] explains, it is recommended that multi-criteria techniques be applied from a practical perspective, that is, from placing ourselves in a paradigm of broad, flexible, and realistic rationality, where the human factor can be incorporated in the search for a better analysis of the problem.

## 5. Conclusions

The complexity of territorial issues is being addressed with widely conflicting and disparate points of view. This complexity arises at different scales, as we have shown in this text, and the sources are overly broad. However, we have tried to treat our work using a basic argument. The physical patterns of development that give rise to the structure of urban space mean that it can be foreseen that the form that urban space will take can be established in advance and that it can be ordered by codes and norms. In other words, there would be certain goals to be set and, since urban planning sets the objectives in the interests of communities, it must be assumed that there must be a general desire to achieve an orderly occupation of geographical space or at least a regulation of the urban process so that it comes as close as possible to the precepts of spatial planning. We believe that it is not accurate, or appropriate according to the term in the title of this paper, to continue to maintain that the forms taken by the urban process are a free play of forces and a multitude of interests and decisions, which can never be effectively guided by any predetermined arrangement set up to optimize occupations and developments. In this extreme way of thinking, cities and expanding metropolitan areas cannot be forms of indiscriminate occupation of territory, as if they were some kinds of final stage of a natural phenomenon.

We have sought an understanding of the process of urban development, rather than an attempt to reach some final conclusions or an imagined or desirable result. The process itself can be described in rational terms and made intelligible. To this end, we have attempted to provide a systematic investigation of the process of change outside cities, illustrating it with maps and data that show the urban morphology and its forms of growth in an area that lends itself energetically to it, such as the coastal areas of the Spanish Mediterranean.

**Author Contributions:** Conceptualization, F.B.G.-J. and S.R.-C.; methodology, F.B.G.-J.; software, F.B.G.-J. and S.R.-C.; validation, S.R.-C. and F.B.G.-J.; formal analysis, F.B.G.-J. and S.R.-C.; investigation, F.B.G.-J.; resources, S.R.-C.; data curation, S.R.-C.; writing—original draft preparation, F.B.G.-J.; writing—review and editing, F.B.G.-J. and S.R.-C.; visualization, S.R.-C.; supervision, F.B.G.-J. All authors have read and agreed to the published version of the manuscript.

**Funding:** This research received no external funding.

**Data Availability Statement:** The data presented in this study are available on request from the corresponding author. The data are not publicly available due to privacy restrictions.

**Conflicts of Interest:** The authors declare no conflicts of interest.

## Appendix A

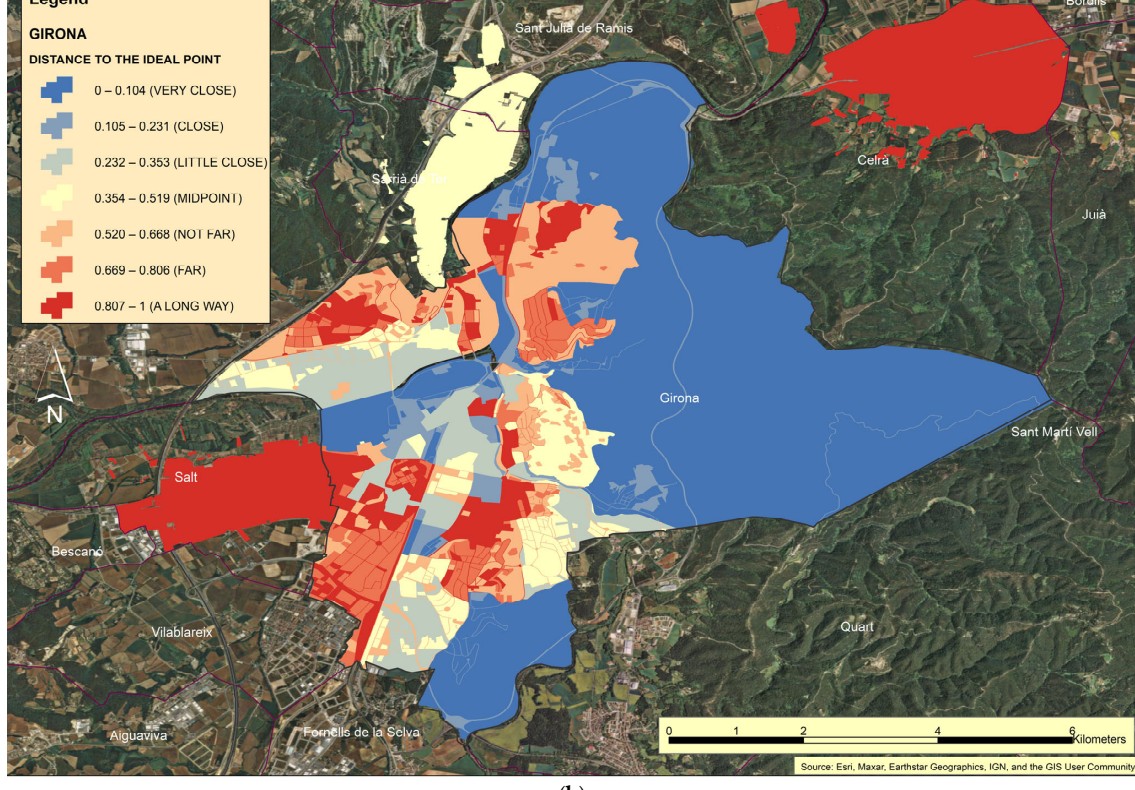

(**a**)

(**b**)

**Figure A1.** *Cont.*

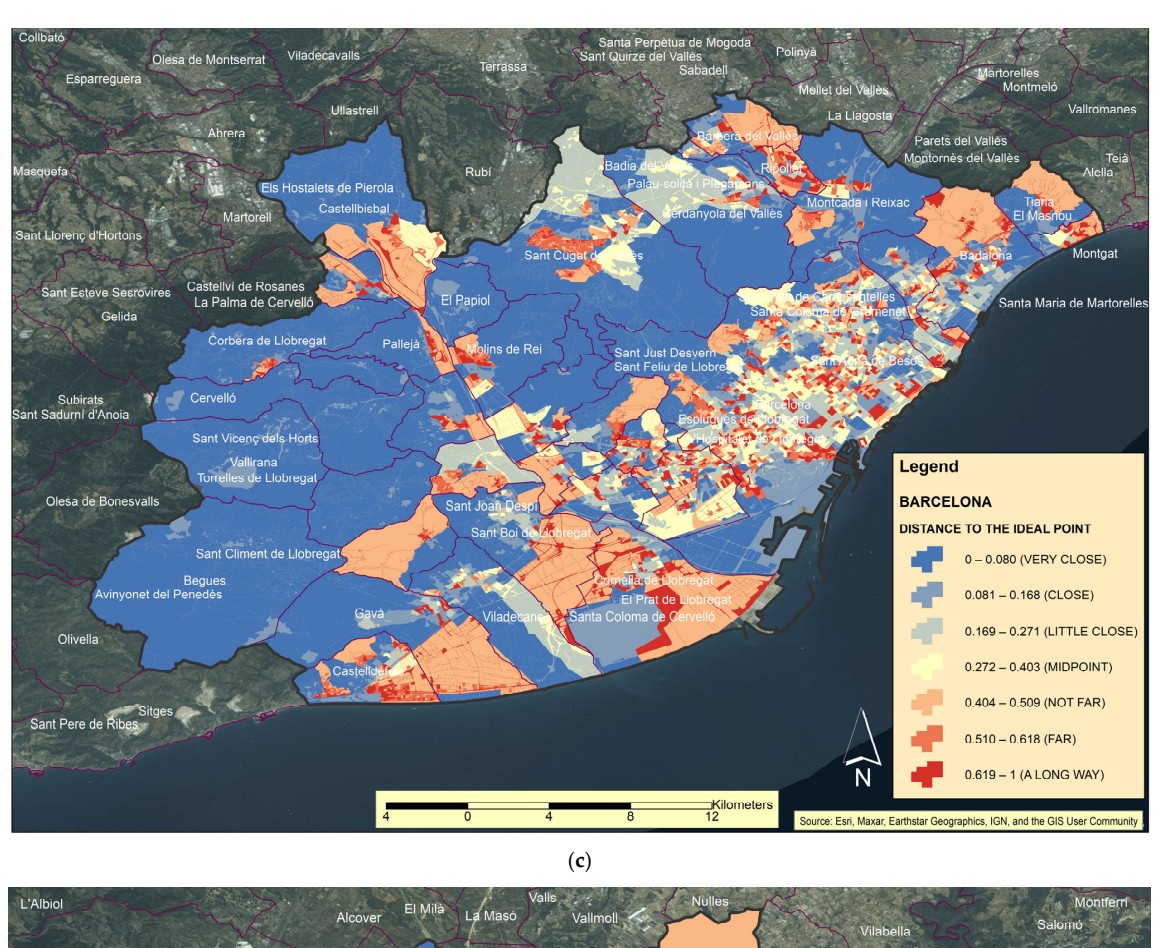

(**c**)

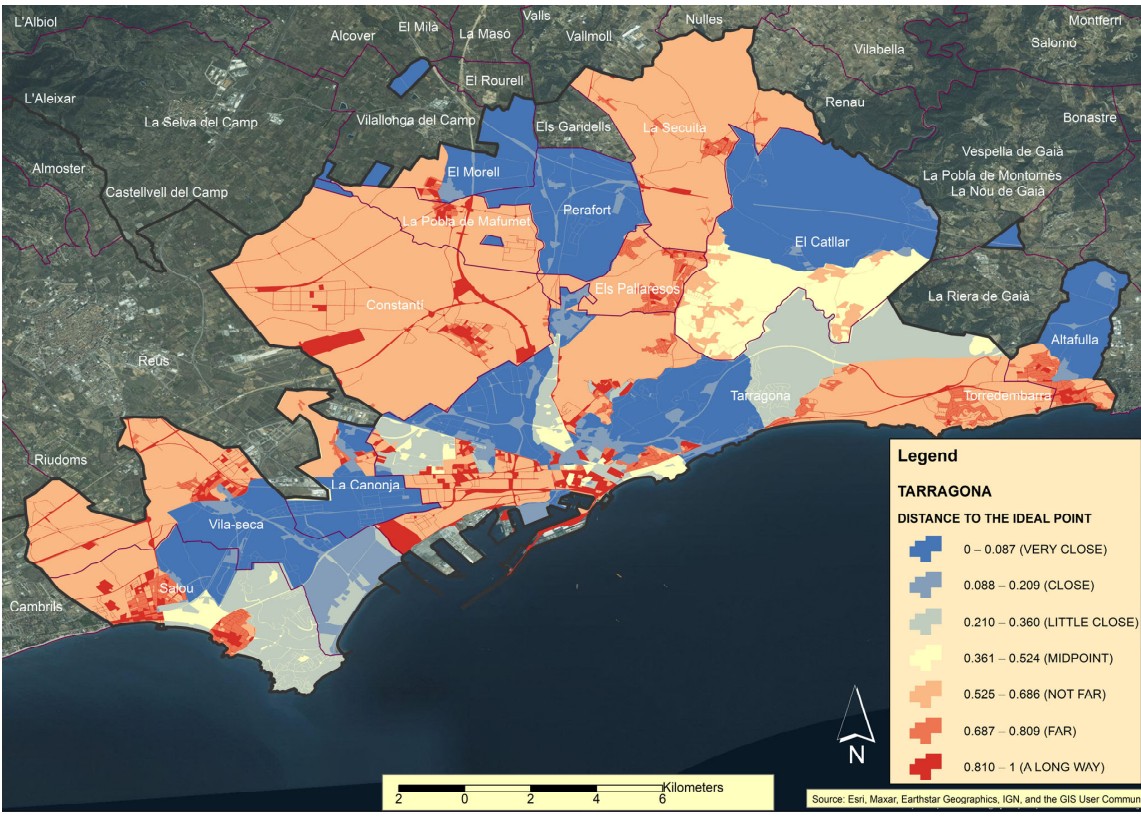

(**d**)

**Figure A1.** *Cont.*

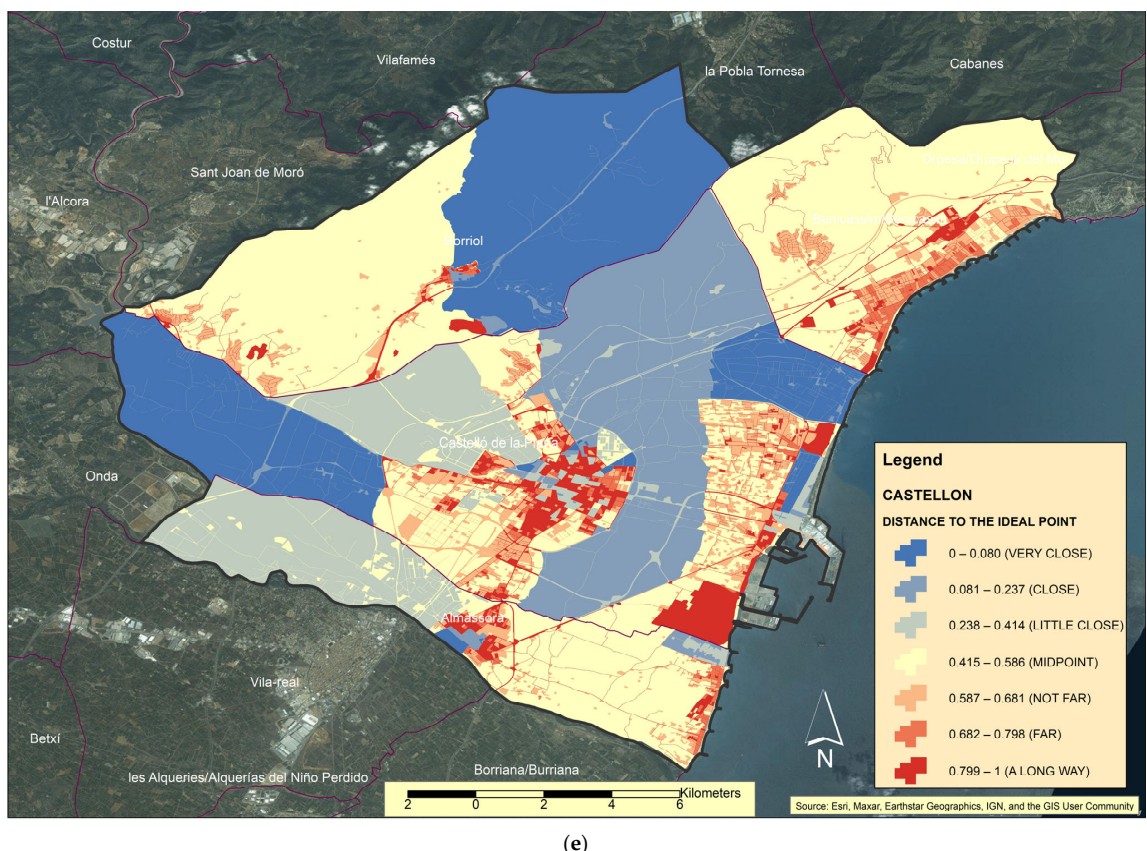

(**e**)

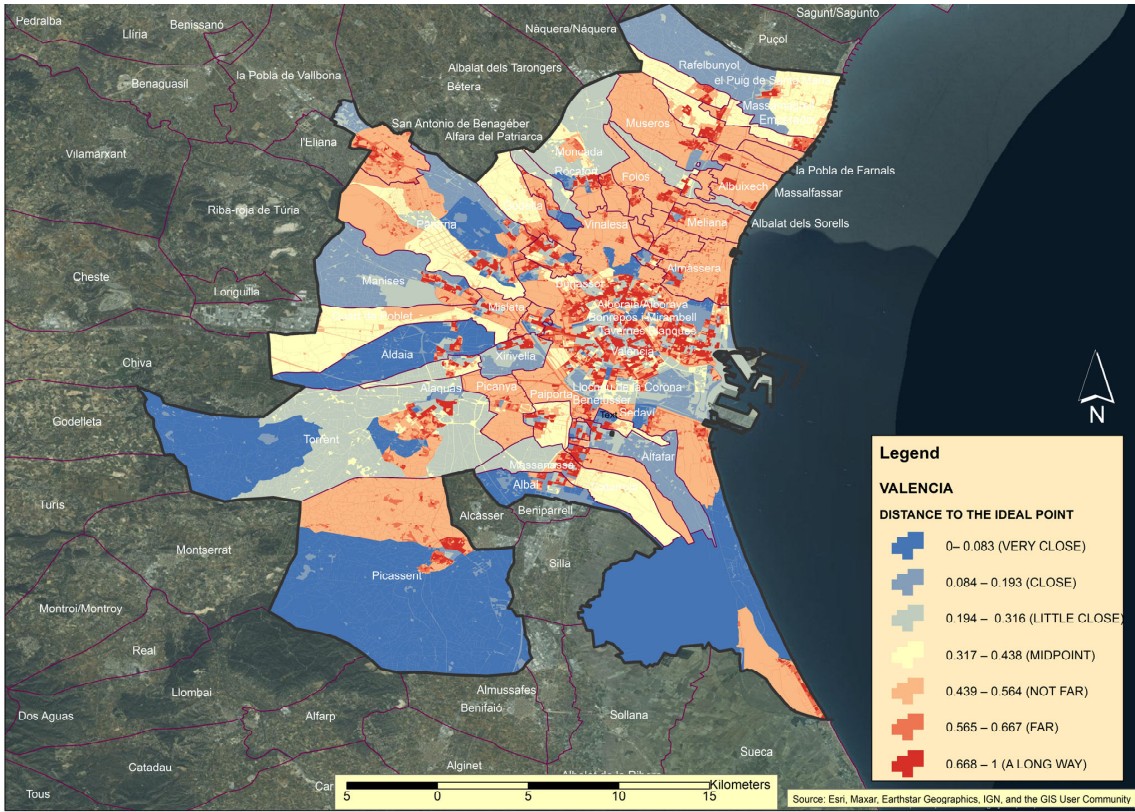

(**f**)

**Figure A1.** *Cont.*

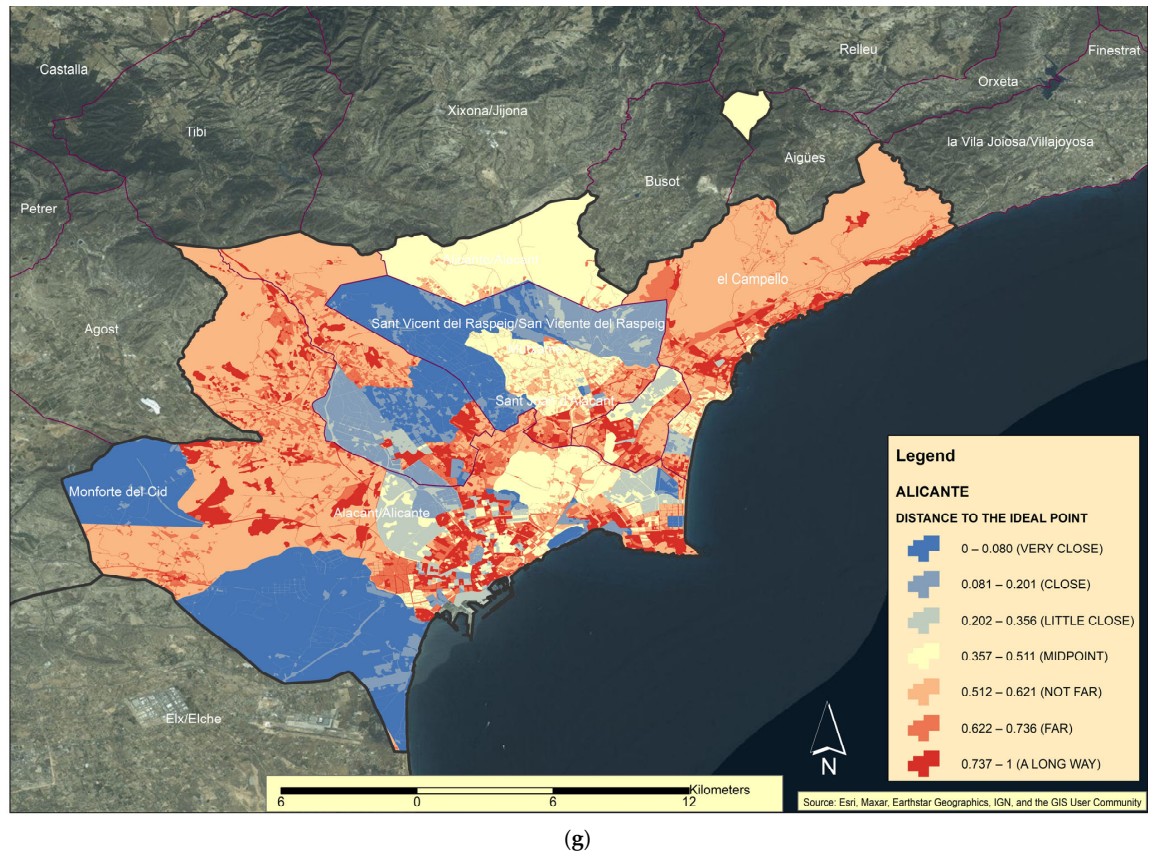

(**g**)

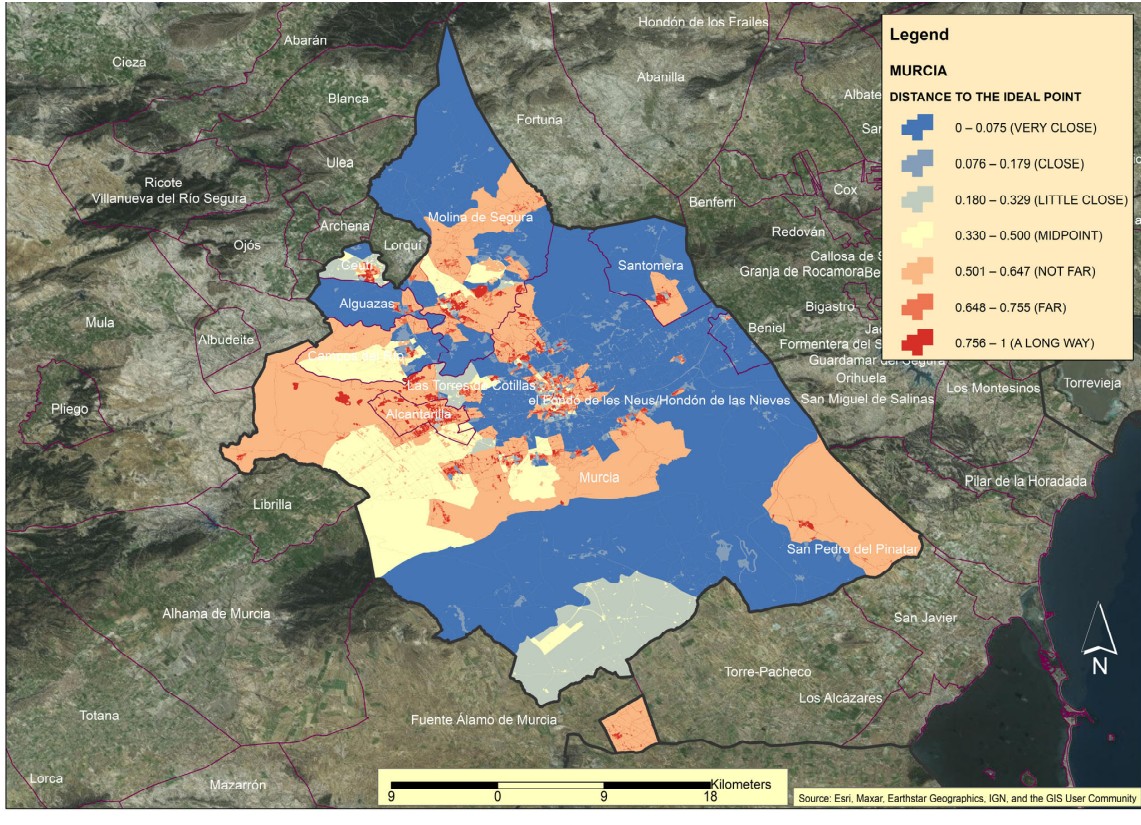

(**h**)

**Figure A1.** *Cont.*

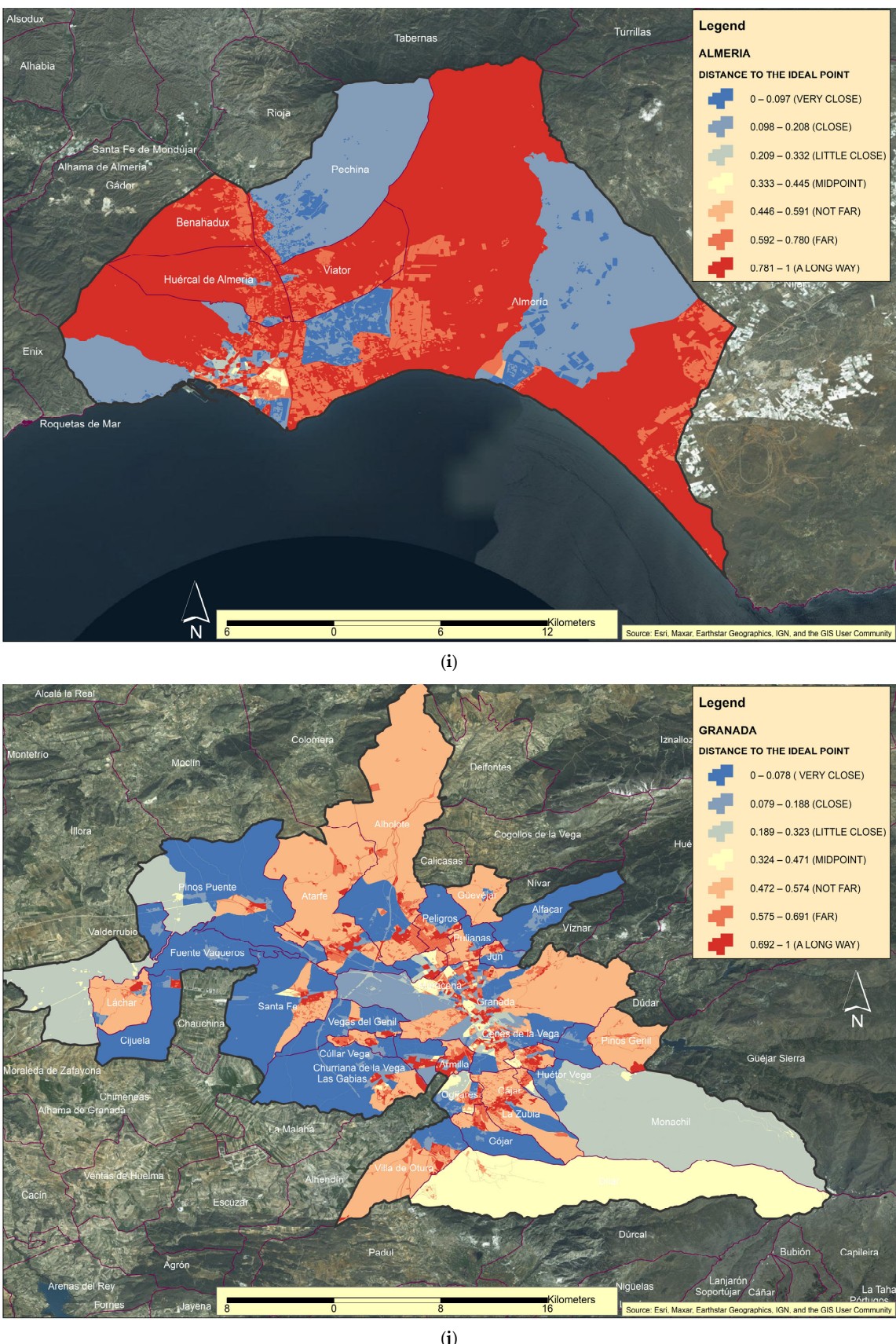

**Figure A1.** *Cont.*

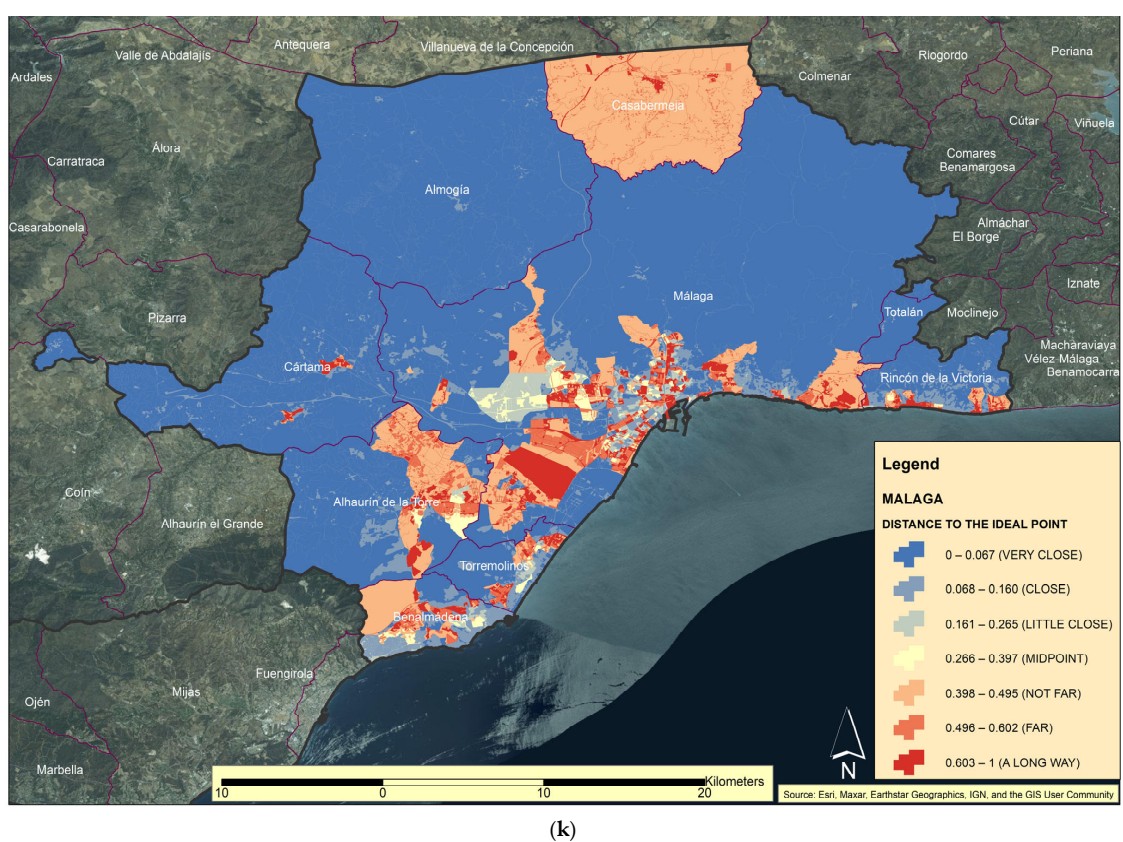

(**k**)

**Figure A1.** Result of the application of the multi-criteria evaluation process in large traditional urban centres and their successive expansions: (**a**) Palma de Mallorca, (**b**) Girona, (**c**) Barcelona, (**d**) Tarragona, (**e**) Castellón, (**f**) Valencia, (**g**) Alicante, (**h**) Murcia, (**i**) Almería, (**j**) Granada, and (**k**) Málaga.

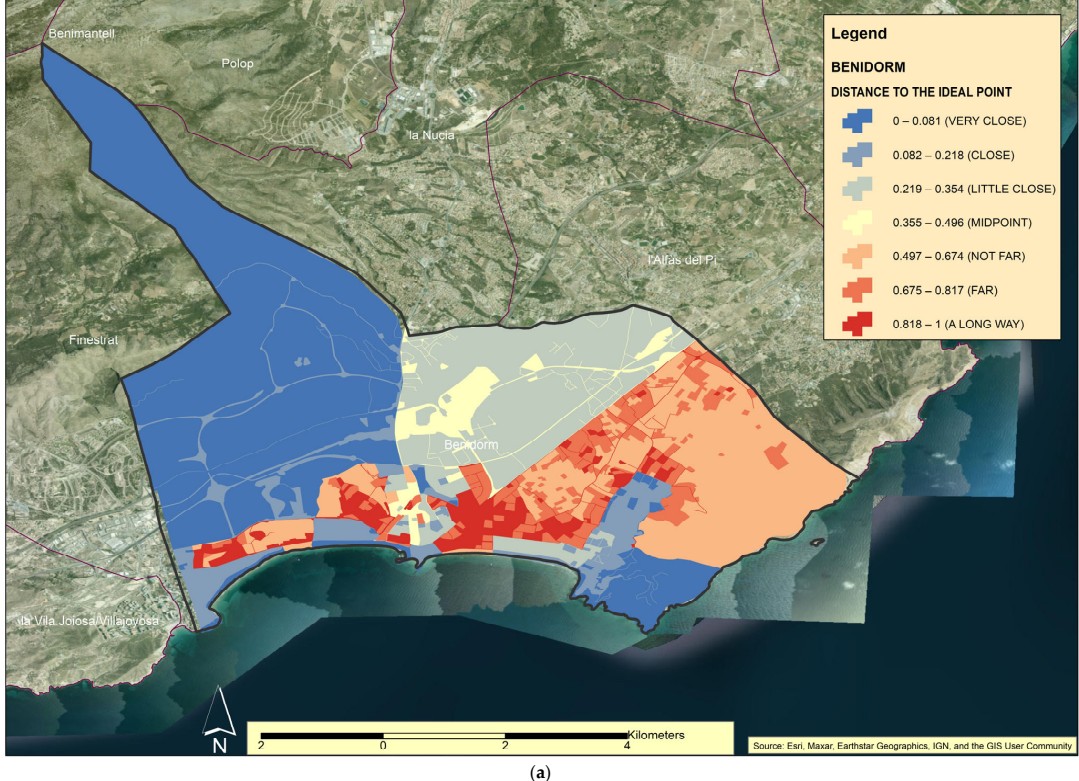

(**a**)

**Figure A2.** *Cont.*

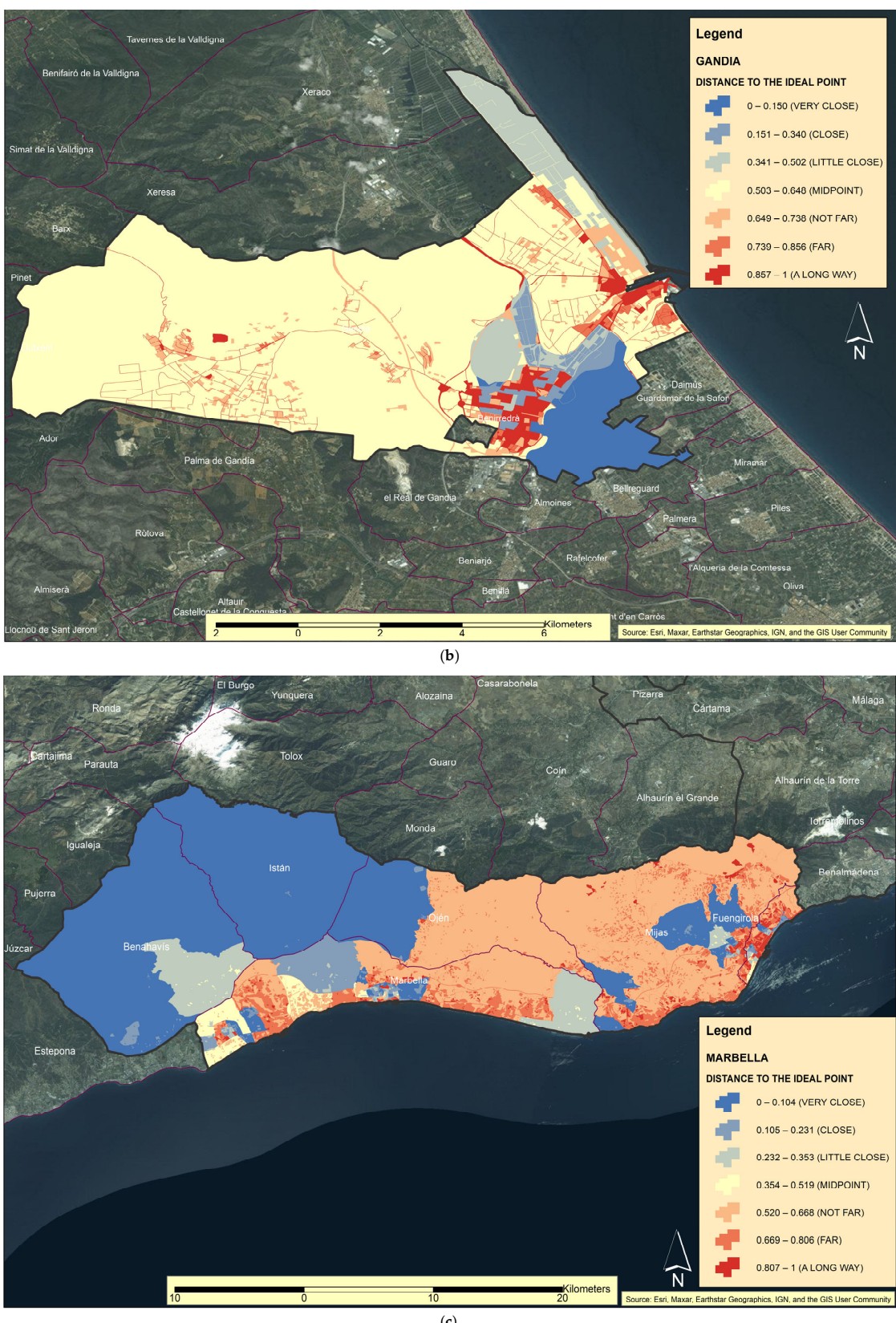

**Figure A2.** *Cont.*

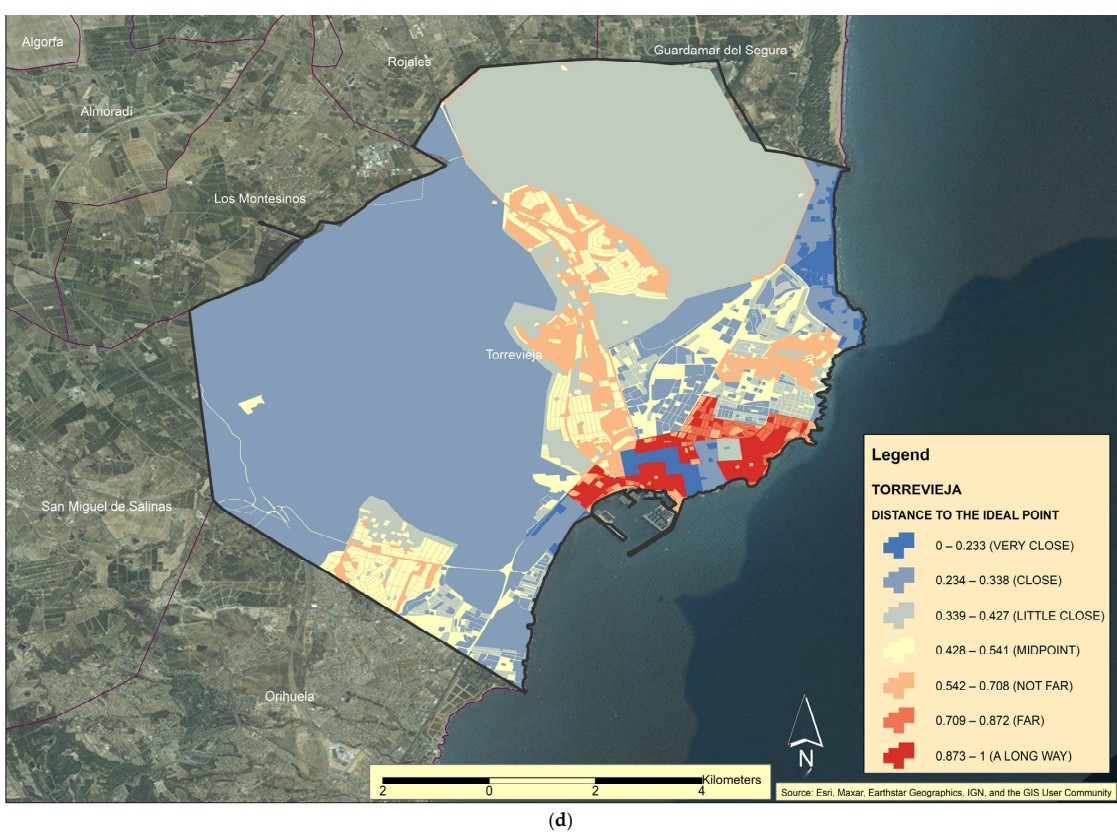

(**d**)

**Figure A2.** Result of the application of the multi-criteria evaluation process in suburban tourist city spaces: (**a**) Benidorm, (**b**) Gandía, (**c**) Marbella, and (**d**) Torrevieja.

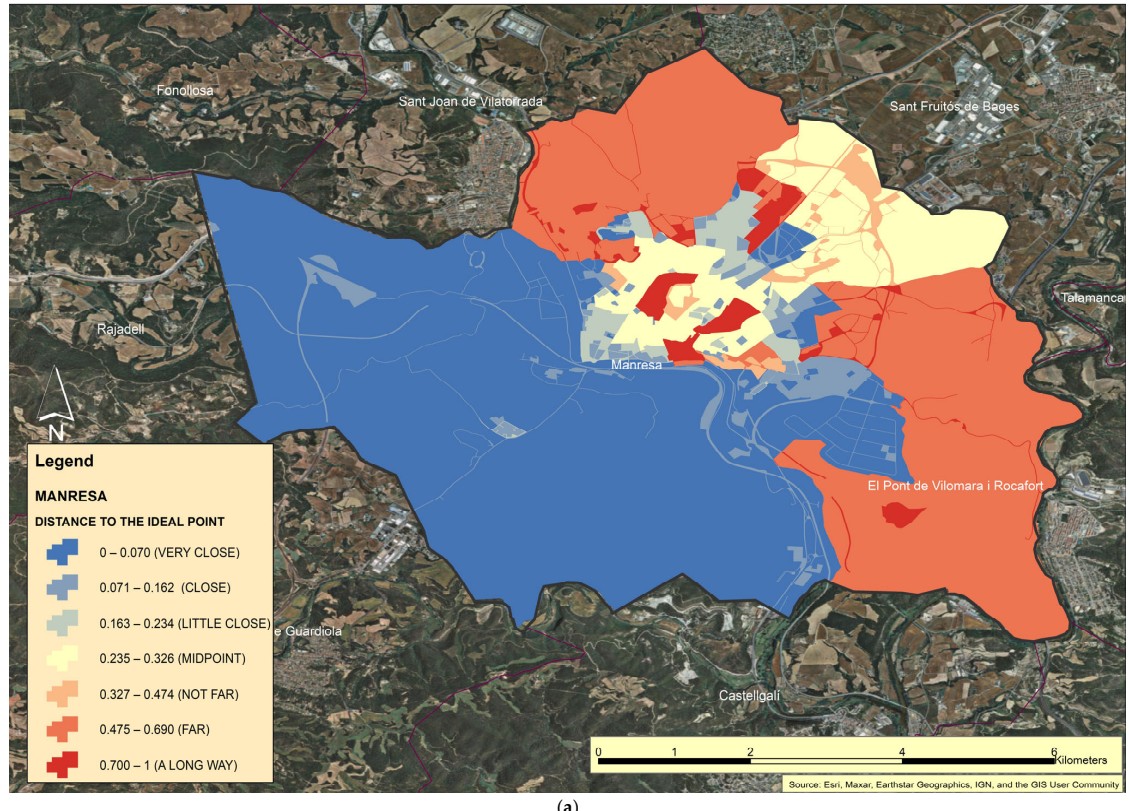

(**a**)

**Figure A3.** *Cont.*

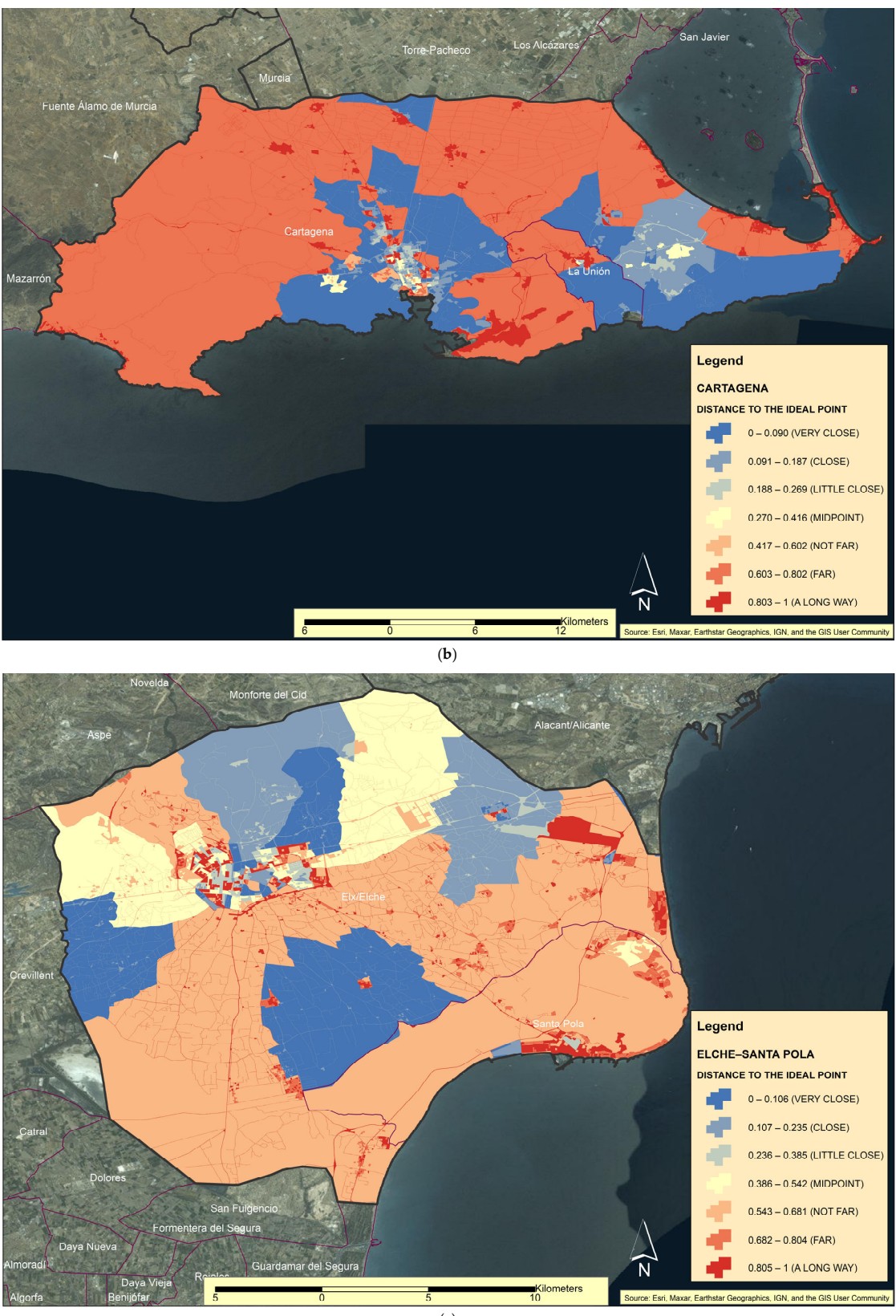

**Figure A3.** *Cont.*

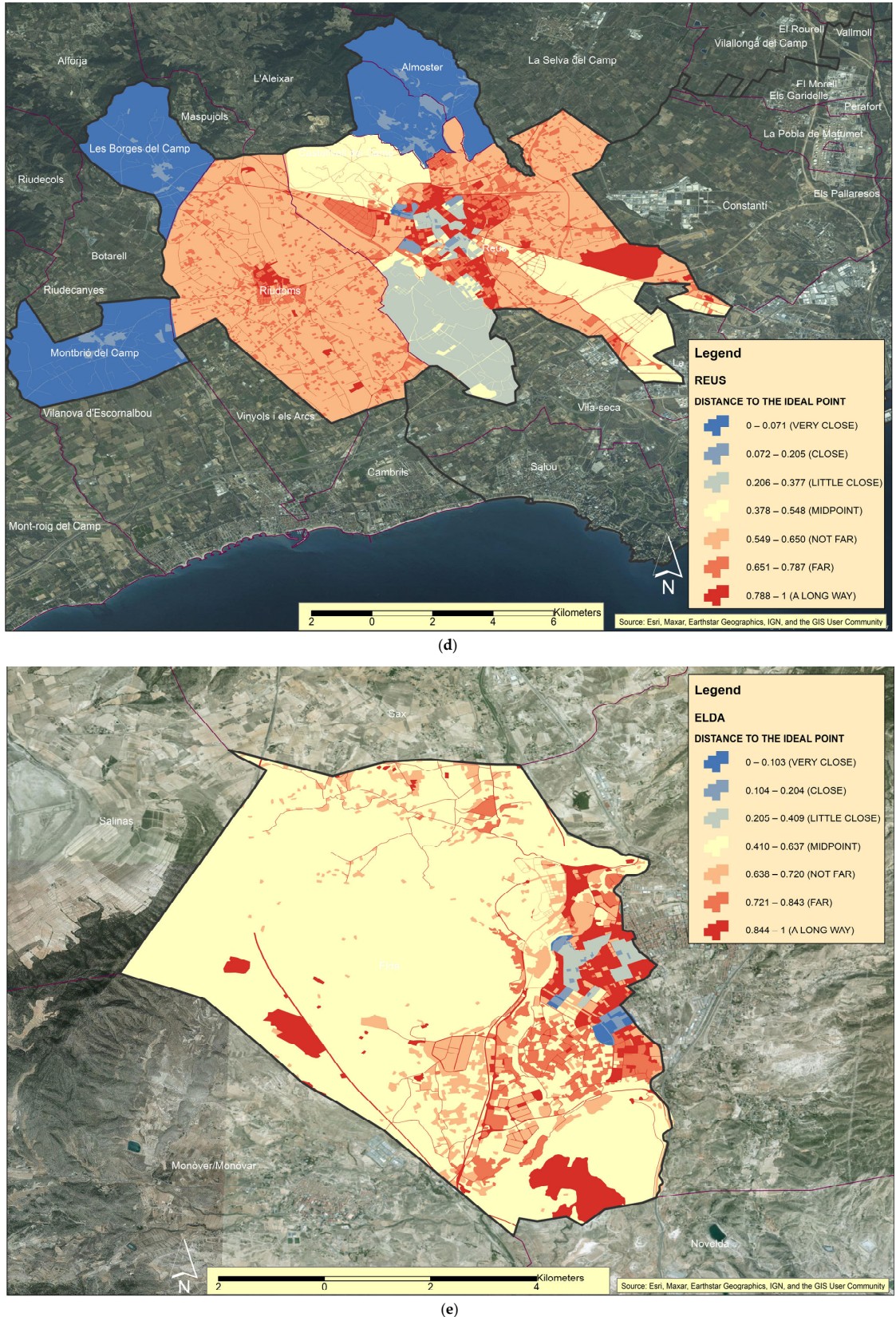

**Figure A3.** Result of the application of the multi-criteria evaluation process in modern and complex suburban spaces: (**a**) Manresa, (**b**) Cartagena, (**c**) Elche–Santa Pola, (**d**) Reus, and (**e**) Elda.

## Appendix B

**Table A1.** Maximum population density values (inhabitants/ha) according to the categories of the distance from the ideal point (Suitability criteria: Very Close, as the best valued, to A Long Way, as the worst rated.

| AREAS | VERY CLOSE | CLOSE | LITTLE CLOSE | MIDPOINT | NOT FAR | FAR | A LONG WAY |
|---|---|---|---|---|---|---|---|
| **Traditional urban centres and their successive extensions** | | | | | | | |
| Murcia | 214 | 477 | 506 | 651 | 833 | 833 | 630 |
| Palma Mallorca | 262 | 416 | 471 | 596 | 821 | 772 | 814 |
| Málaga | 249 | 525 | 716 | 748 | 765 | 1092 | **1483** |
| Granada | 166 | 338 | 483 | 587 | 635 | **669** | **669** |
| Barcelona | 290 | 359 | 644 | 1524 | 1524 | **1531** | **1531** |
| Valencia | 214 | 728 | 728 | 671 | 789 | 894 | 1020 |
| Almería | 218 | 406 | 406 | 443 | 442 | **645** | 390 |
| Alicante | 188 | 432 | 505 | 737 | 624 | 737 | **828** |
| Castellón | 170 | 356 | 412 | 508 | **596** | 357 | **596** |
| Tarragona | 194 | 384 | 384 | 516 | 736 | 618 | 763 |
| Girona | 145 | 261 | 448 | **499** | 251 | 333 | 424 |
| **Suburban tourist city spaces** | | | | | | | |
| Marbella | 231 | 519 | 710 | **712** | **712** | 369 | 369 |
| Benidorm | 66 | 333 | 508 | 625 | 717 | **768** | **768** |
| Torrevieja | 250 | 175 | 175 | 124 | 229 | 229 | 229 |
| Gandía | 237 | 366 | 441 | 368 | 426 | 403 | **701** |
| **Modern and complex suburban spaces** | | | | | | | |
| Cartagena | 189 | 361 | 343 | 461 | 530 | **650** | 234 |
| Elda | 240 | 380 | **544** | **544** | 344 | 318 | 344 |
| Elche-Sta. Pola | 200 | 456 | 508 | 647 | **707** | 553 | 557 |
| Reus | 111 | 364 | 501 | 597 | 679 | 759 | 759 |
| Manresa | 192 | 263 | 253 | 475 | 475 | 99 | 128 |

Data source: Changing Mediterranean Metropolises Around Time (CAT-MED).

**Table A2.** Maximum housing density values (housing/ha) according to the categories of the distance from the ideal point (Suitability criteria: Very Close, as the best valued, to A Long Way, as the worst rated.

| AREAS | VERY CLOSE | CLOSE | LITTLE CLOSE | MIDPOINT | NOT FAR | FAR | A LONG WAY |
|---|---|---|---|---|---|---|---|
| **Traditional urban centres and their successive extensions** | | | | | | | |
| Murcia | 91 | 187 | 238 | 272 | 338 | 338 | **371** |
| Palma Mallorca | 93 | 174 | 194 | 230 | 277 | 295 | **334** |
| Málaga | 85 | 193 | 267 | 306 | 328 | 554 | **708** |
| Granada | 104 | 191 | 261 | 335 | 335 | 381 | **441** |
| Barcelona | 116 | 192 | 249 | 348 | 353 | 599 | **631** |
| Valencia | 103 | 190 | 290 | 326 | 347 | 436 | **512** |
| Almería | 91 | 195 | 228 | 245 | **261** | 327 | 209 |

**Table A2.** *Cont.*

| AREAS | VERY CLOSE | CLOSE | LITTLE CLOSE | MIDPOINT | NOT FAR | FAR | A LONG WAY |
|---|---|---|---|---|---|---|---|
| Alicante | 83 | 191 | 234 | 338 | 303 | 355 | **364** |
| Castellón | 84 | 200 | 215 | **269** | 268 | 162 | 268 |
| Tarragona | 91 | 175 | 208 | 245 | 386 | 347 | **407** |
| Girona | 85 | 159 | 159 | **241** | 117 | 133 | 186 |
| *Suburban tourist city spaces* | | | | | | | |
| Marbella | 84 | 184 | **243** | 348 | 269 | **348** | 174 |
| Benidorm | 91 | 188 | 238 | 343 | 343 | **346** | **346** |
| Torrevieja | 201 | 263 | 263 | 146 | **280** | 264 | **280** |
| Gandía | 115 | 186 | 225 | 154 | 213 | 177 | **371** |
| *Modern and complex suburban spaces* | | | | | | | |
| Cartagena | 77 | 163 | 179 | 213 | 264 | **334** | 88 |
| Elda | 83 | 170 | **262** | **262** | 186 | 167 | 186 |
| Elche-Sta. Pola | 100 | 195 | 216 | 271 | **297** | 224 | 224 |
| Reus | 51 | 194 | 212 | 285 | 285 | **333** | **333** |
| Manresa | 87 | 139 | 133 | **208** | **208** | 82 | 83 |

Data source: INE (National Institute of Statistics). 2022. Government of Spain.

**Table A3.** Maximum values of square meters per inhabitant (m$^2$/inhab.), according to the categories of the distance from the ideal point (Suitability criteria: Very Close, as the best valued, to A Long Way, as the worst rated.

| AREAS | VERY CLOSE | CLOSE | LITTLE CLOSE | MIDPOINT | NOT FAR | FAR | A LONG WAY |
|---|---|---|---|---|---|---|---|
| *Traditional urban centres and their successive extensions* | | | | | | | |
| Murcia | **393** | **393** | 60 | 49 | 23 | 12 | 12 |
| Palma Mallorca | **354** | **354** | 72 | 51 | 46 | 151 | 12 |
| Málaga | **315** | **315** | 71 | 49 | 37 | 24 | 11 |
| Granada | **137** | **137** | 66 | 48 | 23 | 23 | 12 |
| Barcelona | **2536** | 2430 | 2100 | 50 | 36 | 24 | 11 |
| Valencia | **962** | **962** | 61 | 36 | 35 | 18 | 11 |
| Almería | **305** | **305** | 50 | 27 | 27 | 22 | 11 |
| Alicante | **185** | **185** | 75 | 49 | 25 | 23 | 11 |
| Castellón | **137** | **137** | 59 | 37 | 22 | 17 | 11 |
| Tarragona | **121** | **121** | 60 | 47 | 28 | 21 | 10 |
| Girona | **258** | **258** | 57 | 48 | 37 | 20 | 11 |
| *Suburban tourist city spaces* | | | | | | | |
| Marbella | **119** | **119** | 61 | 46 | 34 | 13 | 8 |
| Benidorm | **218** | **218** | 46 | 46 | 40 | 10 | 10 |
| Torrevieja | **48** | **48** | 36 | 36 | 22 | 10 | 10 |
| Gandía | **120** | 38 | 38 | 31 | 21 | 15 | 11 |

**Table A3.** *Cont.*

| AREAS | VERY CLOSE | CLOSE | LITTLE CLOSE | MIDPOINT | NOT FAR | FAR | A LONG WAY |
|---|---|---|---|---|---|---|---|
| **Modern and complex suburban spaces** | | | | | | | |
| Cartagena | **230** | **230** | 57 | 57 | 31 | 22 | 10 |
| Elda | 15 | 15 | 15 | **17** | **17** | 12 | 10 |
| Elche-Sta. Pola | **253** | **253** | 64 | 48 | 23 | 15 | 11 |
| Reus | **121** | **121** | 44 | 44 | 23 | 23 | 10 |
| Manresa | 25 | 35 | 40 | **41** | **41** | 35 | 20 |

Data source: National Geographic Institute. Government of Spain. 2023.

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
