# Peer review of "Spatial Analysis Model for the Evaluation of the Territorial Adequacy of the Urban Process in Coastal Areas"

_land, doi:10.3390/land13010109_

Round 1

Reviewer 1 Report

Comments and Suggestions for Authors

The paper is very well-written, and the methodology appears sound to the best of my knowledge. Moreover, the paper aligns perfectly with the aims and objectives of the LAND journal and is poised to make a significant contribution to the existing literature. I have some minor comments.

The paper, in some instances, uses abbreviations without defining them. This needs correction in the text. Additionally, I highly recommend adding a table of abbreviations at the end of the paper for clarity because you have so many in the paper!

The objectives in Figure 2 and the land use typologies in Table 2 are presented as numbers in brackets, similar to the MDPI citation style. This could potentially confuse readers. Please consider using a different format, such as parentheses, for differentiation.

Regarding Table 5, particularly the sections related to earnings and losses, I require further explanation. The meaning of each column and row in the second and third parts of the table is unclear. For example, I expected identical values in similar cells (e.g., CUF & DUF vs DUF & CUF), but this isn't the case. Additionally, the significance of the diagonal line highlighted in green needs clarification. In the third part of the table (PERCENTAGE OF LOSSES), the sum of all areas does not total 100%, which is confusing.

I am also concerned about the phrase 'Data source: own elaboration.' It is more appropriate to detail the procedures of the elaboration and the sources upon which these are based. Using 'own elaboration' as a data source is unusual for a journal of this calibre.

Finally, the Data Availability Statement listed as 'Not applicable' seems incorrect in this context. Please provide an updated statement.

Author Response

Thank you very much for your feedback. Of course they will be taken care of, in addition to the fact that they are very grateful, because they are made with the intention of improving this manuscript.

Comment number 1.

“The paper, in some instances, uses abbreviations without defining them. This needs correction in the text. Additionally, I highly recommend adding a table of abbreviations at the end of the paper for clarity because you have so many in the paper!”

Answer.

In the text we have corrected all the abbreviations or acronyms that were not clearly defined and we have changed the order of how to express them, so that the abbreviation will always go after its definition.

Comment number 2.

“The objectives in Figure 2 and the land use typologies in Table 2 are presented as numbers in brackets, similar to the MDPI citation style. This could potentially confuse readers. Please consider using a different format, such as parentheses, for differentiation.”

Answer.

It can certainly be confusing, we have replaced brackets with parentheses in both Table 2 and Figure 2. In fact, we've changed it throughout the text.

Comment number 3.

“Regarding Table 5, particularly the sections related to earnings and losses, I require further explanation. The meaning of each column and row in the second and third parts of the table is unclear. For example, I expected identical values in similar cells (e.g., CUF & DUF vs DUF & CUF), but this isn't the case. Additionally, the significance of the diagonal line highlighted in green needs clarification. In the third part of the table (PERCENTAGE OF LOSSES), the sum of all areas does not total 100%, which is confusing.”

Answer.

A number of issues arise here:

“...particularly the sections related to earnings and losses, I require further explanation.”

When a usage type in the column coincides with that same usage type in the row, it means that it has not changed in the period studied (1990-2018) and has been called persistence. For example: CUF has a persistence surface of 51,670 ha. On the diagonal are marked the areas of unchanged use classes. Of the total area of CUF use: 92,209 ha, 51,670 ha are persistences and 40,539 ha are losses in the period studied. CUF use has lost a surface area of 24,960 ha in the period studied in favor of DUF use. And so on, the losses that this type of use has suffered in favor of other types of use are shown. It has suffered a loss in favour of ICT use, which is quantified in 7,031 ha. The sum of the losses from each type of use is quantified in the total losses column: 40,539 ha. In such a way that this amount represents the sum of the losses of the CUF type of use in favor of each of the other types of uses.

The profit column should be understood in a similar way. In this way, we know that the CUF use has been nourished by surfaces of the other types of uses according to the quantities expressed in the corresponding column. And so on with every type of use. For example, the type of use that contributes the most to the growth of the surfaces of the CUF type of use is AAZ, in such a way that a surface 20,944 ha of the AAZ type of use has become CUF in the period studied. 

“The meaning of each column and row in the second and third parts of the table is unclear...”

Although we could do the analysis with the surface data in the first part of the table, we have considered that it could provide more clarity to make the proportions of the areas in two directions, with respect to earnings and with respect to losses. With this we can see the proportions of the different types of uses. In this way we can observe the changes depending on each use. For example, the AAZ type of use is the most dynamic in terms of changes, both in that most of the other uses are nourished by it to grow on their surfaces. For example, the CUF usage type feeds into its growth from the AAZ usage type and does so by 63.60 %. And in a 20.31 % of the DUF type of use. 

The third part of the table makes the same sense, but in the opposite direction to what we have explained before.

The expected values cannot be reciprocated for the following reason. The cross-tabulation matrix assess the total change of land categories according to two pairs of components: net change and swap, as well as gross gains and gross losses. Multiple resolution analysis provides additional information concerning the distances over which land change occurs. These methods enable to focus on the strongest signals of systematic landscape transitions, which is necessary ultimately to link pattern to process. The models extrapolate the change among type of uses from time 1 to time 2. A validation procedure compares the models’ predictions of time 2 to a reference map from time 2. Usually the percent correct is high, which engenders a confidence in the model’s predictive abilities. Closer inspection reveals the high percent correct is attributable primarily to the static state of the use change between time 1 and time 2; if the model predicts persistence, then the model is usually accurate. Furthermore, a null model that predicts no change is often better than a model that predicts change, as the agreement between the reference map of time 1 and the reference map of time 2 is often greater than the agreement between a model’s prediction of time 2 and the reference map of time 2.

“Additionally, the significance of the diagonal line highlighted in green needs clarification.”

A model’s prediction of time 2 has, upon visual inspection, an good fit due to the dominance of persistence. The meaning of persistence, the diagonal of the table, is, as has been said, to show the goodness of the fit of the model and, on the other hand, to assess the dynamics of changes. The higher the persistence, the better fit the model presents in the analysis of each change in land use. Accepting that the adjustment of the model is good, the diagonal acquires another meaning that lies in the assessment of the dynamism of the changes, if the persistence figures are low it means that the type of use affected presents a great dynamism, having a significant volume of losses or gains. Such is the case that a high figure on the diagonal means that it is a use that has remained stable in the period studied. Or, on the contrary, it has undergone major transformations. These are quantified as losses or earnings.

“In the third part of the table (PERCENTAGE OF LOSSES), the sum of all areas does not total 100%, which is confusing.”

At the bottom of the table it has been explained how the percentages have been made. In the case of losses, the value of persistence should not be considered when calculating the percentage. We have checked that the percentages are correct.

Comment number 4.

“I am also concerned about the phrase 'Data source: own elaboration.' It is more appropriate to detail the procedures of the elaboration and the sources upon which these are based. Using 'own elaboration' as a data source is unusual for a journal of this calibre.”

Answer.

We understand that it is not appropriate to make the reference: “Data source: own elaboration”, That's why we've removed it and reference the source directly.

Comment number 5.

“Finally, the Data Availability Statement listed as 'Not applicable' seems incorrect in this context. Please provide an updated statement.”

Answer.

Corrected

Reviewer 2 Report

Comments and Suggestions for Authors

Authors present an interesting paper on the study of the territorial occupation process and the urban  areas developement on the Spanish Mediterranean coast using an objective approach based on Multicriteria Evaluation Techniques in a GIS-based environment.

Considering the methodology proposed (not easy to follow in the entire research), I strongly suggest to insert a flowchart that helps readers to understand the logical framework and the consequential steps.

Please insert the reference to the CAT-MED project (cited three times in the text).

Pay attention to the measure unit: square meter is with m not as capitol letter, see line 835.

To better understand the land use changes, please add a significant zoom-in related to Figure 3, focusing on a small territorial portion (1:5.000-1:10.000 maximum). Moreover, it is important to be coherent in the representation of DUF-CUF-ICT and AGA in all the Figures: so, please, use the same colours from Figure 3 to Figure 8

Comments on the Quality of English Language

I suggest a proofreading to make the sentences more fluent and to remove any possible grammar errors (i.e. line 602 "it")

Author Response

Thank you very much for your feedback. Of course they will be taken care of, in addition to the fact that they are very grateful, because they are made with the intention of improving this manuscript.

Comment number 1.

“Considering the methodology proposed (not easy to follow in the entire research), I strongly suggest to insert a flowchart that helps readers to understand the logical framework and the consequential steps.”

Answer.

We've improved Figure 2 Analysis process and data sources. We believe that the methodological process that has been followed is now clearer.

Comment number 2.

“Please insert the reference to the CAT-MED project (cited three times in the text).”

Answer.

We've fixed that. In the text we have corrected all the abbreviations or acronyms that were not clearly defined and we have changed the order of how to express them, so that the abbreviation will always go after its definition.

Comment number 3.

“Pay attention to the measure unit: square meter is with m not as capitol letter, see line 835.”

Answer.

Corrected

Comment number 4.

“To better understand the land use changes, please add a significant zoom-in related to Figure 3, focusing on a small territorial portion (1:5.000-1:10.000 maximum). Moreover, it is important to be coherent in the representation of DUF-CUF-ICT and AGA in all the Figures: so, please, use the same colours from Figure 3 to Figure 8”

Answer.

We've added an additional explanation in the text of what we want to show with Figure 3. This Figure is complementary to Figures 4 to 8. If, as suggested, the changes in use are shown in more detail, we believe that if we look at the aforementioned figures, it is possible to observe in more detail the result of the changes produced in the period studied. On the other hand, as suggested and to be consistent with the representation, the same colors have been used for the changes in Figures 3 to 8.

Comment number 5.

Comments on the Quality of English Language. 

Answer.

We have revised the text to avoid grammatical errors.

Reviewer 3 Report

Comments and Suggestions for Authors

Dear Authors,

The article is interesting, but I have a few comments.

The first and crucial one. The article is too long, much too long.

In my opinion, it can be written to be no longer than 30 pages. Of course, I understand the authors' intentions to document various issues as best as possible. However, reports or books are better suited for this purpose. The article should be much more compact and focus on the most important issues. Many elements have a regional dimension, and Land is a global journal.

I recommend significantly shortening the article in the documentation and descriptive parts.

The authors point to demographic changes (population growth) in the Spanish Mediterranean region. Another situation - depopulation - occurs in the Atlantic region. It's a pity that more explanations was not devoted to these contrasts.

Sincerely

Author Response

Thank you very much for your feedback. Of course they will be taken care of, in addition to the fact that they are very grateful, because they are made with the intention of improving this manuscript.

Comment number 1.

“In my opinion, it can be written to be no longer than 30 pages. Of course, I understand the authors' intentions to document various issues as best as possible. However, reports or books are better suited for this purpose. The article should be much more compact and focus on the most important issues. Many elements have a regional dimension, and Land is a global journal.”

Answer.

Following the suggestion made by the reviewer regarding the dimensions of the text, we have made a revision so that the main body has been reduced to 30 pages. The remaining 15 pages correspond to References and Appendices A and B, which contain Figures and Tables of results to the process of which representative data is shown in the main text. 

The article has left much more information in the inkwell precisely so as not to exceed a reasonable number of pages for a publication of these characteristics. The topic studied and the data sources used offer a very wide volume of information that may be of interest to other researchers. In this sense, the main intention of the authors has been to document the methodological process and its results in as much detail as possible so that other researchers interested in this topic can follow the complete procedure. And not as in other works, the synthesis of the text leads to the fact that it is not possible to know how to develop this procedure, in such a way that it is impossible to replicate it in other study spaces.

Comment number 2.

“Many elements have a regional dimension, and Land is a global journal.”

Answer.

The scope of study of this work is the Spanish Mediterranean area. Here it has been possible to analyze in sufficient detail how the urban process occupies the territory and how it acquires various urban structures, producing territories of high spatial complexity, being due, above all, to the economic processes linked to tourist activity. It is precisely this link that can be observed in other Mediterranean countries, at least those on the European continent. That is why the proposed methodological procedure allows its application in these other countries, so that the regional dimension ceases to be a regional dimension and becomes a global dimension.

As can be seen, the data sources used are from the European Union and belong to databases published by national bodies with funding from European funds. The work has also dealt with various scales, and if, in any case, the work has been reduced to a regional scope in some parts, it is precisely to observe the detail and nuances of the urban process. Another approach could have been limited to cities, however, we have not carried out the analysis at this scale and we have always done it at the regional level, which, as you know, is made up of provincial areas, which are the parts of the whole. We believe that the territorial structures generated by the urban process cannot be observed from too distant a dimension, since it would not be possible to determine their particularities in the different territories.

Comment number 3.

“The authors point to demographic changes (population growth) in the Spanish Mediterranean region. Another situation - depopulation - occurs in the Atlantic region. It's a pity that more explanations was not devoted to these contrasts.”

Answer.

It is certainly an approach that can be very interesting to discuss, but just before we mentioned that other researchers can use the methodology we have applied to do a comparative or contrast analysis, and this could be an example. As we have discussed in comment 1, the dimensions of the work can exceed the expected content of a scientific article only by addressing the most methodological aspects of the subject that has been its object, if it had been the case of expanding the object of the analysis, we would be deviating from the objectives that govern this article. It would be very interesting to deal with the aspects that he proposes, but the nuances that we can find in the Atlantic regions are different from those that we can observe in the Mediterranean. However, thank you for the idea, because on the basis of it we could undertake other studies that would focus on the demographic component.

Round 2

Reviewer 1 Report

Comments and Suggestions for Authors

Thank you for the systematic revisions. It was difficult to check all the revisions as the resubmitted version did not have "track changes" or highlights. Please consider this for your future papers. 

Author Response

Thank you very much for your comments. We'll consider in future papers.

Best regards.

Reviewer 3 Report

Comments and Suggestions for Authors

Dear Authors,

I have read the authors' answers. I agree that each case study presented in global scientific journals should contribute something more than just a local presentation. However, it can always be done in the most compact way, even if it is a study with a prominent role of methodology.

In this review, I only recommend you to highlight in one or two sentences the fact that the studied part of the Spanish coast is different in terms of demographics from the one that was not studied. And this is certainly another research challenge to be addressed in another article. The Discussion chapter is the appropriate place to do this.

Sincerely

Author Response

Thank you very much for your words. We have included this sentence in our paper as a future line of research.

Best regards.
